# Weakly supervised causal representation learning

**Johann Brehmer**[1]  **Pim de Haan**[1,2]  **Phillip Lippe**[2]  **Taco Cohen**[1]

[1]Qualcomm AI Research[*]
[2]QUVA Lab, University of Amsterdam

## Abstract

Learning high-level causal representations together with a causal model from unstructured low-level data such as pixels is impossible from observational data alone. We prove under mild assumptions that this representation is however identifiable in a weakly supervised setting. This requires a dataset with paired samples before and after random, unknown interventions, but no further labels. We then introduce implicit latent causal models, variational autoencoders that represent causal variables and causal structure without having to optimize an explicit discrete graph structure. On simple image data, including a novel dataset of simulated robotic manipulation, we demonstrate that such models can reliably identify the causal structure and disentangle causal variables.

## 1 INTRODUCTION

The dynamics of many systems can be described in terms of some high-level variables and causal relations between them. Often, these causal variables are not known but only observed in some unstructured, low-level representation, such as the pixels of a camera feed. Learning the causal representations together with the causal structure between them is a challenging problem and may be important for instance for applications in robotics and autonomous driving [1]. Without prior assumptions on the data-generating process or supervision, it is impossible to uniquely identify the causal variables and their causal structure [2, 3].

In this work, we show that a weak form of supervision is sufficient to identify both the causal representations and the structural causal model between them. We consider a setting

---

[*]Qualcomm AI Research is an initiative of Qualcomm Technologies, Inc.

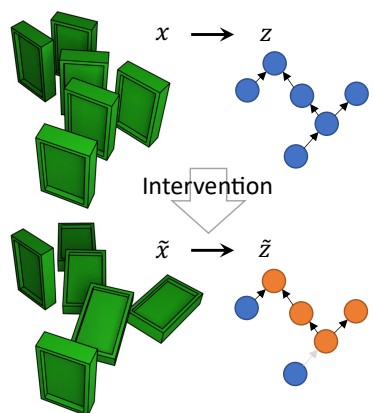

Figure 1: We learn to represent pixels $x$ as causal variables $z$. The bottom shows the effect of intervening on one variable. We prove that variables and causal model can be identified from samples $(x, \tilde{x})$.

in which we have access to data pairs, representing the system before and after a randomly chosen, unknown intervention. Such data may for instance be collected from a video feed of an external agent or demonstrator interacting with a system. Neither labels on the intervention targets nor active control of the interventions are necessary for our identifiability theorem, making this setting useful for offline learning. We prove that with this form of weak supervision, latent causal models (LCMs)—structural causal models (SCMs) together with a decoder from the causal factors to the data space—are identifiable up to a relabelling and elementwise reparameterizations of the causal variables.

We then discuss two practical methods for LCM inference. First, we define explicit latent causal models (ELCMs) as a variational autoencoder (VAE) [4] in which the causal variables are the latent variables and the prior is based on an SCM. While this approach works in simple problems, it can be finicky and is difficult to scale. We trace this to a major challenge in causal representation learning, namely that it is a chicken-and-egg problem: it can be difficult to learn the

*Accepted for the Causal Representation Learning workshop at the 38[th] Conference on Uncertainty in Artificial Intelligence* (UAI CRL 2022).

causal variables when the causal graph is not yet learned, and it is difficult to learn the graph without knowing the variables.

To overcome this optimization difficulty, we introduce a second model class: implicit latent causal models (ILCMs). These models can represent causal structure and variables *without* requiring an explicit, discrete graph representation, which makes gradient-based optimization easier. Nevertheless, these models still contain the causal structure implicitly, and we discuss two algorithms that can extract it after the model is trained. Finally, we demonstrate ILCMs on synthetic datasets, including the new CausalCircuit dataset of a robot arm interacting with a causally connected system of light switches. We show that these models can robustly learn the true causal variables and the causal structure from pixels.

**Related work** Our work builds on the work of Locatello et al. [5] on *disentangled representation learning*. The authors introduce a similar weakly supervised setting where observations are collected before and after unknown interventions. In contrast to our work, however, they focus on disentangled representations, i.e. (conditionally) independent factors of variation with a trivial causal graph, which our work subsumes as a special case. Other relevant works on disentangled representation learning and (nonlinear) independent component analysis include Refs. [6–12].

The problem of *causal representation learning* has been gaining attention lately, see the recent review by Schölkopf et al. [1]. Lu et al. [13] learn causal representations by observing similar causal models in different environments. von Kügelgen et al. [14] use the weakly supervised setting to study self-supervised learning, using a known but nontrivial causal graph between content and style factors. Lippe et al. [15] learn causal representations from time-series data from labelled interventions, assuming that causal effects are not instantaneous but can be temporally resolved. Yang et al. [16] propose to train a VAE with an SCM prior, but require the true causal variables as labels. To the best of our knowledge, our work is the first to provide identifiability guarantees for arbitrary, unknown causal graphs in this weakly supervised setting.

## 2 IDENTIFIABILITY OF LATENT CAUSAL MODELS FROM WEAK SUPERVISION

We now show theoretically that causal variables and mechanisms are identifiable from weak supervision. Here, we only provide informal definitions and assume familiarity with common concepts from causality; see Appendix A.1 for a complete and precise treatment. In Appendix B, we discuss limitations of the setup and possible generalizations.

We describe the causal structure between latent variables

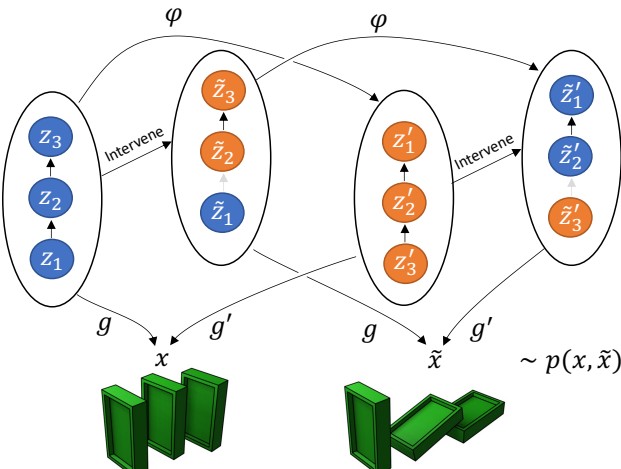

Figure 2: In LCM $\mathcal{M}$, $z_i$ denotes whether the $i$-th stone from the front is standing. Intervening on the second variable, $z_2$, leads to $\tilde{z}$. The decoder $g$ renders $z, \tilde{z}$ as images $x, \tilde{x}$. LCM $\mathcal{M}'$ has an equivalent representation in which $z'_i$ denotes whether the $i$-th stone from the *back* has *fallen*. In Theorem 1, we prove that if and only if two causal models have the same pixel distribution $p(x, \tilde{x})$, there exists an LCM isomorphism $\varphi$: an element-wise reparametrization of the causal variables plus a permutation of the ordering that commutes with interventions and causal mechanisms.

as a Structural Causal Model (SCM). An SCM $\mathcal{C}$ describes the relation between causal variables $z_1, \ldots, z_n$ with domains $\mathcal{Z}_i$ and noise variables $\epsilon_1, \ldots, \epsilon_n$ with domains $\mathcal{E}_i$ along a directed acyclic graph (DAG) $\mathcal{G}(\mathcal{C})$. Causal mechanisms $f_i : \mathcal{E}_i \times \prod_{j \in \mathbf{pa}_i} \mathcal{Z}_j \to \mathcal{Z}_i$ describe how the value of a causal variable is determined from the associated noise variables, as well as the values of its parents in the graph. Finally, an SCM includes a probability measure for the noise variables.

An SCM entails a unique solution $s : \mathcal{E} \to \mathcal{Z}$ defined by successively applying the causal mechanisms. We require the causal mechanisms to be pointwise diffeomorphic, that is, for any value of the parents $z_{\mathbf{pa}_i}$ we have that $f_i(\cdot; z_{\mathbf{pa}_i})$ is a diffeomorphism.[1] Then $s$ is also diffeomorphic and thus noise variables can be uniquely inferred from causal variables. This simplifies the weakly supervised distribution, as the only stochasticity comes from the noise variables and the intervention. The SCM also entails an observational distribution $p_{\mathcal{C}}(z)$ (Markov with respect to the graph of the SCM), which is the pushforward of $p_{\mathcal{E}}$ through the solution.

A perfect, stochastic intervention $(I, (\tilde{f}_i)_{i \in I})$ modifies an SCM by replacing for a subset of the causal variables, called the intervention target set $I \subset \{1, ..., n\}$, the causal mecha-

---

[1]Under some mild smoothness assumptions, any SCM can be brought into this form by elementwise redefinitions of the variables, preserving the observational and interventional distributions, but not the weakly supervised / counterfactual distribution.

nism $f_i$ with a new mechanism $\tilde{f}_i : \mathcal{E}_i \to \mathcal{Z}_i$, which does not depend on the parents. The intervened SCM has a new solution $\tilde{s}_I : \mathcal{E} \to \mathcal{Z}$. We call interventions atomic if the number of targeted variables is one or zero.

We will reason about generative models in a data space $\mathcal{X}$, in which the causal structure is latent. Also including a distribution of interventions, we define LCMs:

**Definition 1** (Latent causal model (LCM)). *A latent causal model $\mathcal{M} = \langle \mathcal{C}, \mathcal{X}, g, \mathcal{I}, p_{\mathcal{I}} \rangle$ consists of*

- *an acyclic, faithful SCM $\mathcal{C}$,*
- *an observation space $\mathcal{X}$,*
- *a decoder $g : \mathcal{Z} \to \mathcal{X}$ that is diffeomorphic onto its image,*
- *a set $\mathcal{I}$ of interventions on $\mathcal{C}$, and*
- *a probability measure $p_{\mathcal{I}}$ over $\mathcal{I}$.*

We define two LCMs as equivalent if all of their components are equal up to a permutation of the causal variables and elementwise diffeomorphic reparameterizations of each variable, see Fig. 2.

**Definition 2** (LCM isomorphism (informal)). *Let $\mathcal{M} = \langle \mathcal{C}, \mathcal{X}, g, \mathcal{I}, p_{\mathcal{I}} \rangle$ and $\mathcal{M}' = \langle \mathcal{C}', \mathcal{X}, g', \mathcal{I}', p'_{\mathcal{I}'} \rangle$ be two LCMs with identical observation space. An LCM isomorphism between them is a graph isomorphism $\psi : \mathcal{G}(\mathcal{C}) \to \mathcal{G}(\mathcal{C}')$ together with elementwise diffeomorphisms for noise and causal variables that tell us how to reparameterize them, such that the structure functions, noise distributions, decoder, intervention set, and intervention distribution of $\mathcal{M}'$ are compatible with the corresponding elements of $\mathcal{M}$ reparameterized through the graph isomorphism and elementwise diffeomorphisms. $\mathcal{M}$ and $\mathcal{M}'$ are equivalent, $\mathcal{M} \sim \mathcal{M}'$, if and only if there is an LCM isomorphism between them.*

Following Ref. [5], we define a generative process of pre and post interventional data:[2]

**Definition 3** (Weakly supervised generative process). *Consider an LCM $\mathcal{M}$ where the underlying SCM has continuous noise spaces $\mathcal{E}_i$, independent probabilities $p_{\mathcal{E}_i}$, and admits a solution $s$. We define the weakly supervised generative process of data pairs $(x, \tilde{x}) \sim p_{\mathcal{M}}^{\mathcal{X}}(x, \tilde{x})$ as follows:*

$$\epsilon \sim p_{\mathcal{E}}, \qquad z = s(\epsilon), \qquad x = g(z),$$
$$I \sim p_{\mathcal{I}}, \quad \forall i \in I, \, \tilde{\epsilon}_i \sim p_{\tilde{\mathcal{E}}_i}, \quad \forall i \notin I, \, \tilde{\epsilon}_i = \epsilon_i,$$
$$\tilde{z} = \tilde{s}_I(\tilde{\epsilon}), \qquad \tilde{x} = g(\tilde{z}). \quad (1)$$

---

[2]This construction is closely related to twinned SCMs [17, Def. 2.17], typically used to compute counterfactual queries $p(\tilde{z}_{\setminus I} | z, \tilde{z}_I)$. We instead focus on the joint distribution of pre-intervention and post-intervention data.

The main theoretical result of this paper is that an LCM $\mathcal{M}$ can be identified from $p(x, \tilde{x})$ up to a relabeling and elementwise transformations of the causal variables:

**Theorem 1** (Identifiability of $\mathbb{R}$-valued LCMs from weak supervision). *Let $\mathcal{M} = \langle \mathcal{C}, \mathcal{X}, g, \mathcal{I}, p_{\mathcal{I}} \rangle$ and $\mathcal{M}' = \langle \mathcal{C}', \mathcal{X}, g', \mathcal{I}', p'_{\mathcal{I}'} \rangle$ be LCMs with the following properties:*

- *The LCMs have an identical observation space $\mathcal{X}$.*
- *The SCMs $\mathcal{C}$ and $\mathcal{C}'$ both consist of $n$ real-valued endogenous causal variables and corresponding exogenous noise variables, i.e. $\mathcal{E}_i = \mathcal{Z}_i = \mathcal{Z}_i' = \mathcal{E}_i' = \mathbb{R}$.*
- *The intervention sets $\mathcal{I}$ and $\mathcal{I}'$ consist of all atomic, perfect interventions, $\mathcal{I} = \{\emptyset, \{z_0\}, \ldots, \{z_n\}\}$ and similar for $\mathcal{I}'$.*
- *The intervention distribution $p_{\mathcal{I}}$ and $p'_{\mathcal{I}'}$ have full support.*

*Then the following two statements are equivalent:*

1. *The LCMs entail equal weakly supervised distributions, $p_{\mathcal{M}}^{\mathcal{X}}(x, \tilde{x}) = p_{\mathcal{M}'}^{\mathcal{X}}(x, \tilde{x})$.*
2. *The LCMs are equivalent, $\mathcal{M} \sim \mathcal{M}'$.*

We prove this result in Appendix A.2.

# 3 IMPLICIT LATENT CAUSAL MODELS

Theorem 1 means that it is possible to learn causal structure from pixel-level data in the weakly supervised setting. Consider a system that is described by an unknown true LCM and assume that we have access to data pairs $(x, \tilde{x})$ sampled from its probability density. Then we can train another LCM with learnable components by maximum likelihood. Assuming sufficient data and perfect optimization, this model's density will match that of the ground-truth LCM. Our identifiability result guarantees that the trained LCM then has the same causal variables and causal structure as the ground truth, up to relabelling.

**Explicit LCMs** The most straightforward neural LCM implementation consists of a variational autoencoder (VAE) [4], in which the latent variables are the causal variables $(z, \tilde{z})$. Such an *explicit* LCM (ELCM) requires a graph representation. In practice we find that learning the causal variables and the causal graph jointly is a challenging optimization problem and prone to local minima in the loss landscape. We discuss these models, show some experimental results, and describe the failure modes in Appendix E.

**Implicit LCMs** To enable causal representation learning in a more robust, scalable way, we propose a second LCM implementation: *Implicit Latent Causal Models* (ILCMs). ILCMs are also variational autoencoders with a causally

structured prior, but unlike ELCMs, they do not require an explicit graph parameterization and are thus easier to train in practice.

**Latents** The latent variables in an ILCMs are *noise encodings*, defined through the inverse solution function as $e = s^{-1}(z)$ and $\tilde{e} = s^{-1}(\tilde{z})$. The pre-intervention noise encoding $e$ is identical to the SCM noise variables. The post-intervention noise encoding $\tilde{e}$ corresponds to the value of the SCM noise variables that would have generated the post-intervention causal variables $\tilde{z}$ under the unintervened SCM mechanisms. ILCMs contain a stochastic encoder $q(e|x)$ and decoder $p(x|e)$ that map data $(x, \tilde{x})$ to noise encodings $(e, \tilde{e})$.

Noise encodings have the convenient property that under an intervention with intervention targets $I$, precisely the components $e_I$ change value: $e_i \neq \tilde{e}_i \Leftrightarrow i \in I$ with probability 1. We prove this property in Appendix A. This means that from noise encodings $e, \tilde{e}$, we can infer interventions easily. We use a simple heuristic intervention encoder that assigns higher intervention probability $q(i \in I|x, \tilde{x})$ to a component $i$ the more this component of the noise encoding changes under interventions:

$$\log q(i \in I|x, \tilde{x}) \sim h\big(\mu_e(x)_i - \mu_e(\tilde{x})_i\big), \qquad (2)$$

where $\mu_e(x)$ is the mean function of the noise encoder $q(e|x)$ and $h$ is a quadratic function with learnable parameters. Both the equality pattern of $e$ under interventions and this heuristic intervention encoder are similar to the ones used for disentangled representation learning in Ref. [5].

**Prior** Given encoders for noise encodings and intervention targets, let us now write down the prior $p(e, \tilde{e}, I)$, which encodes the structure of the weakly supervised setting. The intervention-target prior $p(I)$ and the pre-intervention noise distribution $p(e)$ are given by simple base densities, which we choose as uniform categorical and standard Gaussian, respectively. The post-intervention noise encodings $\tilde{e}$ follow the conditional probability distribution

$$p(\tilde{e}|e, I) = \prod_{i \notin I} \delta(\tilde{e}_i - e_i) \prod_{i \in I} \tilde{p}(\bar{z}_i) \left| \frac{\partial \bar{z}_i}{\partial \tilde{e}_i} \right|, \quad \bar{z}_i = \bar{s}_i(\tilde{e}_i; e_{\setminus i}).$$
$$(3)$$

In the second equality we have parameterized the conditional density $p(\tilde{e}_i|e)$ with a conditional normalizing flow consisting of a learnable diffeomorphic transformation $\tilde{e}_i \mapsto \bar{z}_i = \bar{s}_i(\tilde{e}_i; e_{\setminus i})$ and a base density $\tilde{p}$ on $\bar{z}_i$, which we choose as standard Gaussian.

How does this prior encode causal structure? When intervening on $i$, the map $\bar{s}_i$ transforms $\tilde{e}_i$ into a variable $\bar{z}_i$ that is independent of $e$. This may sound familiar: the causal model gives another such transformation. The noise encoding $\tilde{e}$ is related to the causal variables via the solution function $\tilde{z} = s(\tilde{e})$. The solution function is defined through recursive application of the causal mechanisms, so that $\tilde{z}_i = s_i(\tilde{e}) = f(\tilde{e}_i, s_{\mathbf{pa}_i}(\tilde{e}_{\mathbf{anc}_i}))$. Because we assume perfect stochastic interventions, the output $\tilde{z}_i$ is independent from $\tilde{e}_{\mathbf{anc}_i} = e_{\mathbf{anc}_i}$. Then, it is easy to show (see Appendix C) that for $\mathbb{R}$-valued variables, $\bar{z}_i$ and $\tilde{z}_i$ are related by an elementwise diffeomorphism. Because we desire identifiability up to an elementwise diffeomorphism, we are free to choose $\tilde{z}_i := \bar{z}_i$, and thus $\bar{s}_i(\tilde{e}_i; e_{\setminus i}) := s_i(\tilde{e})$. By learning to transform $\tilde{e}_i$ into $\tilde{z}_i$, we have learned the solution function!

The solution functions can now be also used to infer the unintervened causal variables: $z_i = s_i(e_i; e_{\setminus i})$. We have thus learned a causal model without ever explicitly modelling a graph. Because the solution function only depends on ancestors in the graph, the learned transformation $s_i(e_i; e_{\setminus i})$ should also depend only on ancestors of $i$. As each $s_i$ is constructed to be a diffeomorphism in its first argument, jointly they have a triangular structure and thus a diffeomorphism $s : e \mapsto z$. In practice, however, the learned solution functions may still depend weakly on non-ancestors. Therefore, to ensure that $s$ always forms a diffeomorphism, at some point in training, we test functional dependence to infer ancestral dependence, pick a topological ordering of variables conforming to the ancestry, and parameterize the solution functions $s_i$ to only depend on earlier variables in the ordering.

The final question is how to implement the first terms in Eq. (3), which encode that those noise encodings that are not part of the intervention targets $I$ should not change value under the intervention. We enforce this in the encoder by setting the non-intervention components of $e$ and $\tilde{e}$ to the same value [similar to 5]. In Appendix C this procedure is described in more detail. We will refer to this projective noise encoder as $q(e, \tilde{e}|x, \tilde{x}, I)$.

**Learning** Putting everything together, an ILCM consists of an intervention encoder $q(I|x, \tilde{x})$, a noise encoder $q(e, \tilde{e}|x, \tilde{x}, I)$, a noise decoder $p(x|e)$, and transformations / solution functions $s_i(\cdot; e)$. All of these components are implemented with neural networks and learnable, see Appendix C for details. The lower bound on the joint log likelihood of pre-intervention and post-intervention data is given by

$$\log p(x, \tilde{x}) \geq \mathbb{E}_{I \sim q(I|x, \tilde{x})} \mathbb{E}_{e, \tilde{e} \sim q(e, \tilde{e}|x, \tilde{x}, I)}$$
$$\Big[ \log p(I) + \log p(e) + \log p(\tilde{e}|e, I) - \log q(I|x, \tilde{x})$$
$$- \log q(e, \tilde{e}|x, \tilde{x}, I) + \log p(x|e) + \log p(\tilde{x}|\tilde{e}) \Big]. \quad (4)$$

The model is trained by minimizing the corresponding VAE loss, learning to map low-level data to noise variables (with $q$) and to map noise variables to causal variables (with $s$). The expectation over $I$ is computed via summation, but could alternatively be done with sampling.

**Downstream tasks** Despite the implicit representation of causal structure, we argue that ILCMs let us solve various tasks:

- *Causal representation learning / disentanglement*: ILCMs allow us to map low-level data $x$ to causal variables $z$ by applying the encoder $q$ followed by the solution functions $s$.

- *Intervention inference*: It is also straightforward to infer intervention targets from an observed pair $(x, \tilde{x})$ of pre-intervention and post-intervention data, as this just requires evaluating the intervention-target encoder $q_I(x, \tilde{x})$.

- *Causal discovery / identification*: We propose two methods to infer the causal graphs after training an ILCM. One is to use an off-the-shelf method for causal discovery on the learned representations. Since the ILCM allows us to infer intervention targets, we can use intervention-based algorithms. In this paper, we use ENCO [18], a recent differentiable causal discovery method that exploits interventions to obtain acyclic graphs without requiring constrained optimization. Alternatives to ENCO include DCDI [19] and GIES [20]. Alternatively, we can analyze the causal structure implicitly represented in the learned solution functions $s_i$. We propose a heuristic algorithm that proceeds in three steps. First, it infers the topological order by sorting variables such that $s_i$ only depends on $e_j$ if $z_i$ is after $z_j$ in the topological order. It then iteratively rewrites the solution functions such that they only depend on ancestors in the topological order. Finally, it determines which causal ancestors are direct parents by testing the functional dependence of the causal mechanisms. We describe this algorithm in more detail in Appendix C.

- *Generation of interventions and counterfactuals*: The ILCM entails a generative model for pairs of pre- and post-intervention data. It is straightforward to sample from the joint distribution $p(x, \tilde{x}, I)$, from the conditional $p(x, \tilde{x}|I)$, or from the conditional $p(\tilde{x}|x, I)$.

## 4 EXPERIMENTS

Finally, we demonstrate latent causal models in practice. Here we focus on implicit LCMs; explicit LCMs are demonstrated in similar experiments in Appendix E. We evaluate the causal graphs learned by the ILCM models either with ENCO (ILCM-E) or with the heuristic algorithm described above (ILCM-H).

Since we are to the best of our knowledge the first to study causal representation learning in this weakly supervised setting, we are not aware of any baseline methods designed for this task. We nevertheless compare ILCMs to three other methods. First, we define a *disentanglement VAE* that models the weakly supervised process, but assumes independent

Table 1: Experiment results. We compare our ILCM-E (using ENCO for graph inference) and ILCM-H (with a heuristic for graph inference) to disentanglement VAE (dVAE-E), unstructured $\beta$-VAE, and slot attention baselines. We show the DCI disentanglement score ($D$), the accuracy of intervention inference (Acc), and structural Hamming distance (SHD) between learned and true graph. Best results in bold.

| Dataset | Method | $D$ | Acc | SHD |
|---|---|---|---|---|
| Causal3DIdent | ILCM-E (ours) | **0.99** | **0.98** | **0.00** |
| | ILCM-H (ours) | **0.99** | **0.98** | 0.17 |
| | dVAE-E | 0.82 | **0.98** | 1.67 |
| | $\beta$-VAE | 0.66 | – | – |
| | Slot attention | 0.60 | – | – |
| CausalCircuit | ILCM-E (ours) | **0.97** | **1.00** | **0.00** |
| | ILCM-H (ours) | **0.97** | **1.00** | **0.00** |
| | dVAE-E | 0.34 | **1.00** | 6.00 |
| | $\beta$-VAE | 0.39 | – | – |
| | Slot attention | 0.38 | – | – |

factors of variation rather than a non-trivial causal structure between the variables. This baseline is similar to the method proposed by Ref. [5], but it differs in some implementation details to be more comparable to our ILCM setup. We infer the causal graph between the learned representations with ENCO (dVAE-E). We also compare to an unstructured $\beta$-VAE that treats $x$ and $\tilde{x}$ as i. i. d. and uses a standard Gaussian prior. Finally, we consider a slot attention model [21], which segments the image unsupervisedly into as many objects as there are causal variables. The latent representation associated to each object is considered a learned causal variable.

**Causal3DIdent** We first test ILCMs on an adaptation of the Causal3DIdent dataset [14], which contains images of three-dimensional objects under variable positions and lighting conditions. We consider three causal variables representing object hue, the spotlight hue, and the position of the spotlight. We construct six versions of this dataset, each with a different causal graph, randomly initialized nonlinear structure functions, and heteroskedastic noise. These are mapped to images with a resolution of $64 \times 64$.

ILCMs are able to disentangle the causal variables reliably. The results in Tbl. 1 show that the learned representations are more disentangled than those learned by methods that do not account for causal structure. The ILCM as well as the dVAE baseline can infer interventions with almost perfect accuracy. Unlike the baselines, ILCMs also learn the causal graphs accurately. When using ENCO to extract them, the causal graph is always correct; the heuristic algorithm fails in one instance. In contrast, the acausal dVAE-E baseline does in most cases not find the correct causal graphs.

**CausalCircuit** While Causal3DIdent provides a good test of the ability to disentangle features that materialize in pixel

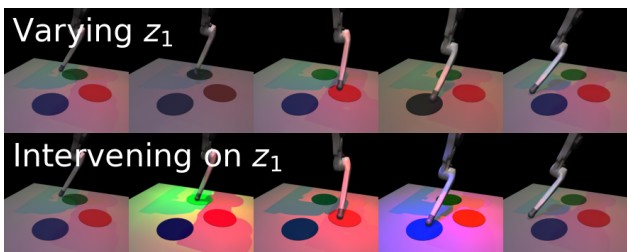

Figure 4: Varying learned causal factors vs. intervening on them. With a trained ILCM, we encode a single CausalCircuit test image (left column). In the top row, we then vary the latent $z_1$ independently, without computing causal effects, and show the corresponding reconstructed images. Only the robot arm position changes, highlighting that we learned a disentangled representation. In the bottom row we instead *intervene* on $z_3$ and observe the causal effects: the robot arm may activate lights, which in turn can affect other lights in the circuit.

space in different ways, like through the position of lights and the color of objects, the underlying causal structure we imposed may feel rather ad-hoc. To explore causal representation learning in a more intuitively causal setting, we introduce a new dataset, which we call CausalCircuit.

The CausalCircuit system consists of a robot arm that can interact with multiple touch-sensitive lights. The lights are connected with a stochastic circuit: a light is more likely to be on if its button is pressed or if its parent lights are on. The robot arm itself can be seen as part of the causal system. Concretely, we consider the

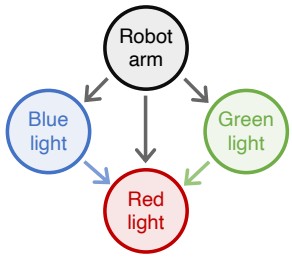

Figure 3: CausalCircuit graph.

causal graph shown in Fig. 3. This system is observed from a fixed-position camera, and we generate samples in $512 \times 512 \times 3$ resolution with MuJoCo [22], see Appendix D for more details.

ILCMs are again able to disentangle the causal variables reliably and better than the acausal baselines, see Tbl. 1. As shown in Appendix D, the slot attention model fails because the lights have no limited spatial extent and thus are not well represented by segments of the image. Interventions are identified with high accuracy. The model also correctly learns the causal graph shown in Fig. 3, both when extracted with ENCO and with our heuristic algorithm. In Fig. 4 we demonstrate how ILCMs let us infer and manipulate causal factors and reason about interventions.

By studying variations of this dataset, we tested the limitations of our method. We find that it works reliably only as long as the causal variables are continuous (that is, when we model the lights with a continuous intensity). As soon as we consider discrete states, the assumptions of our identifiability theorem are violated and the model has difficulty disentangling these variables.

## 5 DISCUSSION

What makes a variable causal? One school of thought is that it that causal variables are those aspects of a system that can be intervened upon [23]. Following this logic, we find it interesting to ask: can we uniquely determine the causal variables underlying a system just by observing the effect of interventions?

In this work we have found a partial answer to this question: we have shown in theory and practice that under certain assumptions, causal variables and their causal structure are identifiable from low-level representations like the pixels of a camera feed if the system is observed before and after random, unlabeled interventions. Our identifiability theorem extends the results by Locatello et al. [5] from independent factors of variation (trivial causal graphs) to arbitrary causal graphs.

LCMs can be implemented as a variational autoencoder. However, a straightforward, explicit parameterization of the causal structure requires simultaneously learning the variables and the causal graph. We found that leads to challenging optimization problems, especially when scaling to larger systems. As a more robust alternative, we introduced implicit latent causal models (ILCMs), which parameterize causal structure without requiring an explicit graph representation. We also discussed two algorithms for extracting the learned causal mechanisms and graph after training.

In first experiments, we demonstrated that ILCMs let us reliably disentangle causal factors, identify causal graphs, and infer interventions from unstructured pixel data. For these experiments, we introduced the new CausalCircuit dataset, which consists of images of a robot arm interacting with connected switches and lights.

The setting we consider is motivated by a potentially useful scenario: learning causal structure from passive observations of an agent (or demonstrator) interacting with a causal system. However, it is currently far from practical. Our identifiability result relies on a number of assumptions, including that interventions are stochastic and perfect, that all atomic interventions may be observed, and that the causal variables are real-valued. We discuss these requirements and their potential relaxation in Appendix B. Similarly, our practical implementation has so far been restricted to simplified datasets with relatively few, continuous causal variables. While more work will be required to make latent causal models applicable to real-world settings, we believe that our results demonstrate that causal representation learning is possible without explicit labels.

**Acknowledgments** We want to thank Thomas Kipf, Dominik Neuenfeld, and Frank Rösler for useful discussions and Gabriele Cesa, Yang Yang, and Yunfan Zhang for helping with our experiments.

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

# A IDENTIFIABILITY RESULT

## A.1 DEFINITIONS

Here we define objects and relations that were not formally defined in the main body of the paper, but are necessary to make Thm. 1 precise and to prove it.

We use the following notation:

- $[n] = \{1, ..., n\}$
- $\mathbf{pa}_i^{\mathcal{C}} \subseteq [n]$ the set of parent nodes of node $i$ in graph $\mathcal{G}(\mathcal{C})$.
- $\mathbf{desc}_i^{\mathcal{C}} \subseteq [n]$ the set of descendant nodes of node $i$ in graph $\mathcal{G}(\mathcal{C})$, excluding $i$ itself.
- $\mathbf{anc}_i^{\mathcal{C}} \subseteq [n]$ the set of ancestor nodes of node $i$ in graph $\mathcal{G}(\mathcal{C})$, excluding $i$ itself.
- $\mathbf{nonanc}_i^{\mathcal{C}} = [n] \setminus (\mathbf{anc}_i^{\mathcal{C}} \cup \{i\})$ the set of non-ancestor nodes of node $i$ in graph $\mathcal{G}(\mathcal{C})$, excluding $i$ itself.
- Given measure $p$ on space $A$ and measurable function $f : A \to B$, $f_* p$ is the push-forward measure on $B$.

We describe causal structure with SCMs.

**Definition 4** (Structural causal model (SCM)). *An SCM is a tuple $\mathcal{C} = \langle \mathcal{Z}, \mathcal{E}, F, p_{\mathcal{E}} \rangle$ consisting of the following:*

- *domains $\mathcal{Z} = \mathcal{Z}_1 \times \cdots \times \mathcal{Z}_n$ of causal (endogenous) variables $z_1, \ldots, z_n$;*
- *domains $\mathcal{E} = \mathcal{E}_1 \times \cdots \times \mathcal{E}_n$ of noise (exogenous) variables $\epsilon_1, \ldots, \epsilon_n$;*
- *a directed acyclic graph $\mathcal{G}(\mathcal{C})$, whose nodes are the causal variables and edges represent causal relations between the variables;*
- *causal mechanisms $F = \{f_1, \ldots, f_n\}$ with $f_i : \mathcal{E}_i \times \prod_{j \in \mathbf{pa}_i} \mathcal{Z}_j \to \mathcal{Z}_i$; and*
- *a probability measure $p_{\mathcal{E}}(\epsilon) = p_{\mathcal{E}_1}(\epsilon_1) p_{\mathcal{E}_2}(\epsilon_2) \ldots p_{\mathcal{E}_n}(\epsilon_n)$ with full support that admits a continuous density.*

*Additionally, we assume that $\forall i, \forall z_{\mathbf{pa}_i}, f_i(\cdot, z_{\mathbf{pa}_i}) : \mathcal{E}_i \to \mathcal{Z}_i$ is a diffeomorphism.*

We will need to reason about vectors being "equal up to permutation and elementwise reparameterizations". We formalize this in the following definition:

**Definition 5** ($\psi$-diagonal). *Let $\psi : [n] \to [n]$ be a bijection (that is, a permutation). Let $\varphi : \prod_{i=1}^{n} X_i \to \prod_{i=1}^{n} Y_i$ be a function between product spaces. Then $\varphi$ is $\psi$-diagonal if there exist functions, called components, $\varphi_i : X_i \to Y_{\psi(i)}$ such that $\forall i, \forall x, \varphi(x_1, ..., x_i, ..., x_n)_{\psi(i)} = \varphi_i(x_i)$.*

This lets us define isomorphisms between SCMs:

**Definition 6** (Isomorphism of SCMs). *Let $\mathcal{C} = \langle \mathcal{Z}, \mathcal{E}, F, p_{\mathcal{E}} \rangle$ and $\mathcal{C}' = \langle \mathcal{Z}', \mathcal{E}', F', p'_{\mathcal{E}} \rangle$ be SCMs. An isomorphism $\varphi : \mathcal{C} \to \mathcal{C}'$ consists of*

1. *a graph isomorphism $\psi : \mathcal{G}(\mathcal{C}) \to \mathcal{G}(\mathcal{C}')$ that tells us how to identify corresponding variables in the two models and which preserves parents: $\mathbf{pa}_{\psi(i)}^{\mathcal{C}'} = \psi(\mathbf{pa}_i^{\mathcal{C}})$ and*

2. *$\psi$-diagonal diffeomorphisms for noise and endogenous variables that tell us how to reparameterize them $\varphi_{\mathcal{E}} : \mathcal{E} \to \mathcal{E}'$ and $\varphi_{\mathcal{Z}} : \mathcal{Z} \to \mathcal{Z}'$, where $\varphi_{\mathcal{E}}$ must be measure preserving $p_{\mathcal{E}'} = \varphi_{\mathcal{E}*} p_{\mathcal{E}}$. For notational simplicity, we will drop the subscript in $\varphi_{\mathcal{Z}}$ and use the symbol $\varphi$ to refer both to the SCM isomorphism and the noise isomorphism.*

*The elementwise diffeomorphisms are required to make the following diagrams commute $\forall i, i' = \psi(i)$:*

$$
\begin{array}{ccc}
\mathcal{Z}_{\mathbf{pa}_i} \times \mathcal{E}_i & \xrightarrow{(\varphi_{\mathbf{pa}_i}, \varphi_{\mathcal{E},i})} & \mathcal{Z}'_{\mathbf{pa}'_{i'}} \times \mathcal{E}'_{i'} \\
\downarrow{f_i} & & \downarrow{f'_{i'}} \\
\mathcal{Z}_i & \xrightarrow{\varphi_i} & \mathcal{Z}'_{i'}
\end{array}
\tag{5}
$$

*Intuitively, this says that if we apply a causal mechanism $f_i$ and then reparameterize the causal variable $i$ using $\varphi_i$, we*

*get the same thing as first reparameterizing the parents and noise variable of variable $i$, and then applying the causal mechanism $f'_{i'}$.*

To reason about interventions, we equip SCMs with intervention distributions in the following definition.

**Definition 7** (Intervention structural causal model (ISCM)). *An intervention structural causal model (ISCM) is a tuple $\mathcal{D} = \langle \mathcal{C}, \mathcal{I}, p_{\mathcal{I}} \rangle$ of*

1. *an acyclic SCM $\mathcal{C} = \langle \mathcal{Z}, \mathcal{E}, F, p_{\mathcal{E}} \rangle$ that admits a faithful distribution, meaning that conditional independence of causal variables $z$ implies $d$-separation [24].*

2. *a set $\mathcal{I}$ of interventions on $\mathcal{C}$, where each intervention $(I, (\tilde{f}_i)_{i \in I}) \in \mathcal{I}$ consist of*

   (a) *a subset $I \subset \{1, ..., n\}$ of the causal variables, called the intervention target set, and*

   (b) *for each $i \in I$, a new causal mechanism $\tilde{f}_i : \mathcal{E}_i \to \mathcal{Z}_i$ which replaces the original mechanism and which does not depend on the parents.*

   *We define intervention set $\mathcal{I}$ to be atomic if the number of targeted variables is one or zero.*

3. *a probability measure $p_{\mathcal{I}}$ over $\mathcal{I}$.*

We can extend the notion of isomorphism from SCMs to ISCMs.

**Definition 8** (Isomorphism of ISCMs). *Let $\mathcal{D} = \langle \mathcal{C}, \mathcal{I}, p_{\mathcal{I}} \rangle$ and $\mathcal{D}' = \langle \mathcal{C}', \mathcal{I}', p_{\mathcal{I}'} \rangle$ be ISCMs. An ISCM isomorphism is an SCM isomorphism $\varphi : \mathcal{C} \to \mathcal{C}'$ with underlying graph isomorphism $\psi : \mathcal{G}(\mathcal{C}) \to \mathcal{G}(\mathcal{C}')$ and a $\psi$-diagonal diffeomorphism $\tilde{\varphi}_{\mathcal{E}} : \mathcal{E} \to \mathcal{E}$ such that*

- *the graph isomorphism $\psi$ induces a bijection of intervention sets*

$$\psi_{\mathcal{I}} : \mathcal{I} \to \mathcal{I}' : (I, (\tilde{f}_i)_{i \in I}) \mapsto (\psi(I), (\tilde{f}'_{i'})_{i' \in \psi(I)})$$

- *for each intervention $(I, (\tilde{f}_i)_{i \in I}) \in \mathcal{I}$, and each intervened on variable $i \in I$, the following diagram commutes:*

$$\begin{CD} \mathcal{E}_i @>{\tilde{\varphi}_{\mathcal{E},i}}>> \mathcal{E}'_{\psi(i)} \\ @V{\tilde{f}_i}VV @VV{\tilde{f}'_{\psi(i)}}V \\ \mathcal{Z}_i @>{\varphi_i}>> \mathcal{Z}'_{\psi(i)} \end{CD} \quad (6)$$

- *$\tilde{\varphi}_{\mathcal{E}}$ is measure preserving, i.e. $p_{\mathcal{E}'} = (\tilde{\varphi}_{\mathcal{E}})_* p_{\mathcal{E}}$.*

- *the bijection $\psi_{\mathcal{I}} : \mathcal{I} \to \mathcal{I}'$ preserves the distribution over interventions: $\psi_* p_{\mathcal{I}} = p'_{\mathcal{I}'}$.*

Latent Causal Models (LCMs), defined in Def. 1, add a map to the data space to an ILCM. We can lift ISCM isomorphisms to LCM isomorphisms by requiring that these decoders must respect the ISCM isomorphism.

**Definition 9** (Isomorphism of LCMs). *Let $\mathcal{M} = \langle \mathcal{C}, \mathcal{X}, g, \mathcal{I}, p_{\mathcal{I}} \rangle$ and $\mathcal{M}' = \langle \mathcal{C}', \mathcal{X}, g', \mathcal{I}', p'_{\mathcal{I}'} \rangle$ be LCMs with identical observation space $\mathcal{X} = \mathcal{X}'$. An LCM isomorphism of LCM is an ISCM isomorphism $\varphi : \mathcal{D} \to \mathcal{D}'$ such that the decoders respect the SCM isomorphism, so this diagram must commute:*

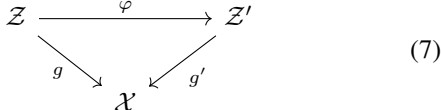

$$(7)$$

*Remark* 1. By defining objects and isomorphisms, we have defined a groupoid of SCMs, a groupoid of ISCMs and a groupoid of LCMs, as the isomorphisms are composed and inverted in an obvious way.

**Definition 10** (Equivalence). *We call two SCMs, ISCMs, or LCMs equivalent if an isomorphism exists between them.*

Informally, two SCMs, ISCMs, or LCMs are equivalent if there is a $\psi$-diagonal map between their causal variables (i.e. the causal variables are equal up to permutation and elementwise diffeomorphisms), there is a $\psi$-diagonal map between their noise encodings, and all other structure (decoders, intervention sets, intervention distributions) is compatible with these reparameterizations.

Next, we define the solution function of an SCM or ISCM, which maps from noise variables to causal variables by repeatedly applying the causal mechanisms.

**Definition 11** (Solution). *Given an ISCM $\mathcal{D} = \langle \mathcal{C}, \mathcal{I}, p_{\mathcal{I}} \rangle$, the solution function $s : \mathcal{E} \to \mathcal{Z}$ is the unique function such that for all $i \in [n]$, the following diagram commutes [17]*

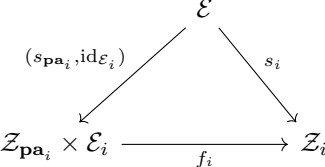

*In equations, we have that $s(\epsilon)_i = f(\epsilon_i; s(\epsilon_{\mathbf{pa}_i}))$. Similarly, intervention $(I, (\tilde{f}_i)_{i \in I}) \in \mathcal{I}$ yields a solution function $\tilde{s}_I : \mathcal{E} \to \mathcal{Z}$ with the modified causal mechanisms.*

For example, with two variables with $z_1 \to z_2$, the solution is given by:

$$s : \mathcal{E} \to \mathcal{Z} : \begin{pmatrix} \epsilon_1 \\ \epsilon_2 \end{pmatrix} \mapsto \begin{pmatrix} z_1 \\ z_2 \end{pmatrix} = \begin{pmatrix} f_1(\epsilon_1) \\ f_2(\epsilon_2, f_1(\epsilon_1)) \end{pmatrix}.$$

Since we require causal mechanisms to be pointwise diffeomorphic, the solution function is a diffeomorphism as well.

Pushing the noise distribution of an SCM through the solution function finally gives us the (observable) distribution

$$\begin{array}{ccc}
\epsilon,\tilde{\epsilon} & \xrightarrow{\varphi_{\mathcal{E}},\tilde{\varphi}_{\mathcal{E}}} & \epsilon',\tilde{\epsilon}' \\
{\scriptstyle s,\tilde{s}_I}\downarrow & & \downarrow{\scriptstyle s',\tilde{s}'_{I'}} \\
z,\tilde{z} & \xrightarrow{\varphi_{\mathcal{Z}},\varphi_{\mathcal{Z}}} & z,\tilde{z}' \\
{\scriptstyle g,g}\downarrow & & \downarrow{\scriptstyle g',g'} \\
x,\tilde{x} & =\!=\!=\!= & x,\tilde{x}
\end{array}$$

Figure 5: An illustration of the spaces and maps in our definitions and proof. When LCMs $\mathcal{M}, \mathcal{M}'$ are isomorphic, all squares in the diagram should commute. Additionally, all maps should preserve the weakly supervised distributions on the variables and all horizontal maps should be $\psi$-diagonal. Note that the latent variables $(\epsilon, z)$ can differ up to a diffeomorphism, but the $x$ variables are actually observed, so must be identically equal. From that equality, the other horizontal maps are uniquely defined.

entailed by an SCM or ISCM. In an ISCM or LCM we can define several other (observable or interventional) distributions.

**Definition 12** (Distributions). *Given an LCM $\mathcal{M} = \langle \mathcal{C}, \mathcal{X}, g, \mathcal{I}, p_{\mathcal{I}} \rangle$, we have the following generative process:*

$$I \sim p_{\mathcal{I}},$$
$$\begin{aligned}
\epsilon &\sim p_{\mathcal{E}}, & \tilde{\epsilon} &\sim \tilde{p}_{\tilde{\mathcal{E}}}(\tilde{\epsilon} \mid \epsilon, I), \\
z &= s(\epsilon), & \tilde{z} &= \tilde{s}_I(\tilde{\epsilon}), \\
x &= g(z), & \tilde{x} &= g(\tilde{z}), \\
e &= s^{-1}(z), & \tilde{e} &= s^{-1}(\tilde{z}),
\end{aligned} \tag{8}$$

*where $p(\tilde{\epsilon}_i \mid \epsilon_i, i \in I) = p_{\mathcal{E}_i}(\tilde{\epsilon}_i)$ and $p(\tilde{\epsilon}_i \mid \epsilon_i, i \notin I) = \delta(\tilde{\epsilon}_i \mid \epsilon_i)$ is the Dirac measure.*

*Then we define the following weakly supervised distributions:*

- *The weakly supervised noise distribution with interventions: $p_{\mathcal{C}}^{\mathcal{E},\mathcal{I}}(\epsilon, \tilde{\epsilon}, I)$.*
- *The weakly supervised causal distribution with interventions: $p_{\mathcal{C}}^{\mathcal{Z},\mathcal{I}}(z, \tilde{z}, I)$.*
- *The weakly supervised observational distribution with interventions: $p_{\mathcal{M}}^{\mathcal{X},\mathcal{I}}(x, \tilde{x}, I)$.*

*These distributions are given by appropriate pushforwards of the noise distributions through the transformations in Eq. (8).*

*By marginalizing over I, we get $p_{\mathcal{C}}^{\mathcal{E}}, p_{\mathcal{C}}^{\mathcal{Z}}, p_{\mathcal{C}}^{e}, p_{\mathcal{M}}^{\mathcal{X}}$ respectively.*

The relationships between all the maps can be found in Fig. 5.

## A.2 IDENTIFIABILITY PROOF

First, we prove two auxiliary lemmata.

**Lemma 1.** *Let $f : [0,1] \to [0,1]$ be differentiable and Lebesgue measure preserving. Then either $f(x) = x$ or $f(x) = 1 - x$.*

*Proof.* We follow the proof from Stack Exchange user zhw [25]. Let $\lambda$ be the Lebesgue measure. Measure preservation means that for any measurable subset $U \subseteq [0,1]$, $\lambda(U) = \lambda(f^{-1}(U))$.

First, note that $f$ is surjective, because otherwise the image of $f$ is a proper subinterval $[a,b] \subsetneq [0,1]$ and $\lambda(f^{-1}([a,b])) = \lambda([0,1]) = 1 > \lambda([a,b]) = b - a$, which contradicts measure-preservation.

Define the open ball $B(x,r) = \{y \in [0,1] \mid |y - x| < r\}$. Suppose that $f'(0) = 0$ for some $x \in [0,1]$. Then there exists an $r > 0$ such that $f(B(x,r)) \subseteq B(f(x), r/4)$, and thus $B(x,r) \subseteq f^{-1}(B(f(x), r/4))$. Therefore, $r \leq \lambda(B(x,r)) \leq \lambda(f^{-1}(B(f(x), r/4)))$, while $\lambda(B(f(x), r/4)) \leq 2 \cdot r/4 = r/2$, contradicting measure preservation. Hence $f'(x) \neq 0$ on $[0,1]$.

By the Darboux theorem, $f'$ is either strictly positive or strictly negative on the interval and thus $f$ is either strictly increasing or decreasing and thus a bijection. Assume that it is strictly increasing, then $\forall x \in [0,1], x = \lambda([0,x]) = \lambda(f^{-1}(f([0,x]))) = \lambda(f([0,x])) = f(x) - f(0) = f(x)$. Similarly, if it is strictly decreasing, we find $f(x) = 1 - x$. $\square$

**Lemma 2.** *Let $A = C = \mathbb{R}$ and $B = \mathbb{R}^n$. Let $f : A \times B \to C$ be differentiable. Define differentiable measures $p_A$ on $A$ and $p_C$ on $C$. Let $\forall b \in B, f(\cdot, b) : A \to C$ be measure-preserving. Then $f$ is constant in $B$.*

*Proof.* Let $P_A : A \to [0,1], P_C : C \to [0,1]$ be the diffeomorphic cumulative density functions. Then $P_A^{-1}$ and $P_C^{-1}$ are measure-preserving maps from the uniform distribution on $[0,1]$. Now write $g : [0,1] \times B \to [0,1] : (z,b) \mapsto P_C(f(P_A^{-1}(z), b))$ such that this diagram of measure-preserving differentiable maps commutes:

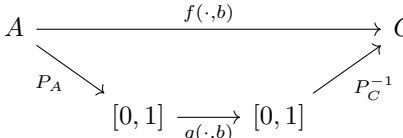

Then $g$ is differentiable and $\forall b \in B$ measure-preserving $[0,1] \to [0,1]$. By the previous Lemma 1, the only differentiable measure-preserving functions $[0,1] \to [0,1]$ are $\mathrm{id}$ and $1 - \mathrm{id}$. As $g$ is continuous in $B$, it can not vary between $\mathrm{id}$ and $1 - \mathrm{id}$ and thus $g$, and consequently $f$ are constant in $B$. $\square$

We can interpret this lemma in terms of statistical independence. Starting from a product measure on $A \times B$, the requirements of the lemma correspond to $a \perp\!\!\!\perp b$ and $c \perp\!\!\!\perp b$. The

lemma thus defines a sense in which for real-valued variables, statistical independence implies functional independence (the converse is always true).

Now in the remainder of this subsection, we prove the main theorem.

**Theorem 1** (Identifiability of $\mathbb{R}$-valued LCMs from weak supervision)**.** *Let* $\mathcal{M} = \langle \mathcal{C}, \mathcal{X}, g, \mathcal{I}, p_{\mathcal{I}} \rangle$ *and* $\mathcal{M}' = \langle \mathcal{C}', \mathcal{X}, g', \mathcal{I}', p'_{\mathcal{I}'} \rangle$ *be LCMs with the following properties:*

- *The SCMs $\mathcal{C}$ and $\mathcal{C}'$ both consist of $n$ real-valued endogeneous variables, i. e. $\mathcal{E}_i = \mathcal{Z}_i = \mathcal{Z}'_i = \mathcal{E}'_i = \mathbb{R}$.*

- *The intervention sets $\mathcal{I}$ and $\mathcal{I}'$ consist of the empty intervention and all atomic interventions, $\mathcal{I} = \{\emptyset, \{z_0\}, \ldots, \{z_n\}\}$ and similar for $\mathcal{I}'$.*

- *The intervention distribution $p_{\mathcal{I}}$ and $p'_{\mathcal{I}'}$ have full support.*

*Then the following two statements are equivalent:*

1. *The weakly supervised distributions entailed by the LCMs are equal, $p_{\mathcal{M}}(x, \tilde{x}) = p_{\mathcal{M}'}(x, \tilde{x})$.*

2. *The LCMs are equivalent, $\mathcal{M} \sim \mathcal{M}'$.*

*Proof.* "(2) $\Rightarrow$ (1)": If the LCMs are equivalent, then the fact that $\varphi_{\mathcal{E}}$ and $\tilde{\varphi}_{\mathcal{E}}$ are measure preserving and that diagrams (5) and (6) commute, implies that $p_{\mathcal{C}'}^{\mathcal{Z}'} = (\varphi_{\mathcal{Z}}, \varphi_{\mathcal{Z}})_* p_{\mathcal{C}}^{\mathcal{Z}}$. Then because diagram (7) commutes, the weakly supervised distributions coincide, $p_{\mathcal{M}'}^{\mathcal{X}} = p_{\mathcal{M}}^{\mathcal{X}}$.

"(1) $\Rightarrow$ (2)": Conversely, if the weakly supervised distributions coincide, $p_{\mathcal{M}'}^{\mathcal{X}} = p_{\mathcal{M}}^{\mathcal{X}}$, the images of $g : \mathcal{Z} \to \mathcal{X}, g' : \mathcal{Z}' \to \mathcal{X}$ coincide,

$$\varphi = g'^{-1} \circ g : \mathcal{Z} \to \mathcal{Z}' \tag{9}$$

is a diffeomorphism, and $\phi$ preserves the weakly supervised distribution over causal variables: $p_{\mathcal{C}'}^{\mathcal{Z}'} = (\varphi, \varphi)_* p_{\mathcal{C}}^{\mathcal{Z}}$.

LCM equivalence then follows from showing that $\varphi : \mathcal{D} \to \mathcal{D}'$ is an ISCM isomorphism, where $\mathcal{D} = \langle \mathcal{C}, \mathcal{I}, p_{\mathcal{I}} \rangle$ and $\mathcal{D}' = \langle \mathcal{C}', \mathcal{I}', p'_{\mathcal{I}'} \rangle$ be the ISCMs inherent to $\mathcal{M}$ and $\mathcal{M}'$. We show this in the following steps:

1. For each intervention $I$ in $\mathcal{D}$, there is a corresponding intervention $I'$ in $\mathcal{D}'$, given by a permutation $\psi : [n] \to [n]$, such that $\varphi$ preserves the interventional distribution.

2. The diffeomorphism $\varphi$ is $\psi$-diagonal.

3. The permutation $\psi$ preserved the ancestry structure of graphs $\mathcal{G}(\mathcal{C})$ and $\mathcal{G}(\mathcal{C}')$.

4. The diffeomorphism $\varphi_{\mathcal{E}} : \mathcal{E} \to \mathcal{E}$ of noise variables is $\psi$-diagonal.

5. The causal mechanisms are compatible with $\varphi$.

**Step 1: Interventions preserved**   Remember that the diffeomorphism $\varphi : \mathcal{Z} \to \mathcal{Z}'$ is such that $p_{\mathcal{C}'}^{\mathcal{Z}'} = (\varphi, \varphi)_* p_{\mathcal{C}}^{\mathcal{Z}}$. For atomic interventions $I \neq J \in \mathcal{I}$, consider the intersection of the supports of the weakly supervised distribution for interventions on $I$ and $J$: $U = \operatorname{supp} p_{\mathcal{C}}^{\mathcal{Z}, \mathcal{I}}(z, \tilde{z} \mid I) \cap \operatorname{supp} p_{\mathcal{C}}^{\mathcal{Z}, \mathcal{I}}(z, \tilde{z} \mid J) \subset \mathcal{Z} \times \mathcal{Z}$. Note that $U$ has zero measure in $p_{\mathcal{C}}^{\mathcal{Z}, \mathcal{I}}(U \mid I) = p_{\mathcal{C}}^{\mathcal{Z}, \mathcal{I}}(U \mid J) = 0$. The distribution is thus a discrete mixture on $(z, \tilde{z})$ of non-overlapping distributions.

The diffeormorphism $(\varphi, \varphi)$ must map between these mixtures. Thus there exists a bijection $\psi : \mathcal{I} \to \mathcal{I}'$, also inducing a permutation $\psi : [n] \to [n]$, such that

$$p_{\mathcal{C}'}^{\mathcal{Z}', \mathcal{I}'} = (\varphi, \varphi, \psi)_* p_{\mathcal{C}}^{\mathcal{Z}, \mathcal{I}} .$$

**Step 2: $\varphi$ is $\psi$-diagonal**   This measure preservation lets us define two equal distributions on $\mathcal{Z} \times \widetilde{\mathcal{Z}}' \times \mathcal{I}$, namely $(\operatorname{id}_{\mathcal{Z}}, \varphi, \operatorname{id}_{\mathcal{I}})_* p_{\mathcal{C}}^{\mathcal{Z}, \mathcal{I}}$ and $(\varphi^{-1}, \operatorname{id}_{\widetilde{\mathcal{Z}}'}, \psi^{-1})_* p_{\mathcal{C}'}^{\mathcal{Z}', \mathcal{I}'}$. In particular, these must then have equal conditionals $p(\tilde{z}' \mid z, I)$. Thus, for any $U \subseteq \widetilde{\mathcal{Z}}', z \in \mathcal{Z}, I \in \mathcal{I}$,

$$p_{\mathcal{C}'}^{\mathcal{Z}', \mathcal{I}'}(\tilde{z}' \in U \mid \varphi(z), \psi(I)) = p_{\mathcal{C}}^{\mathcal{Z}, \mathcal{I}}(\tilde{z} \in \varphi^{-1}(U) \mid z, I)$$

The conditional probability $p_{\mathcal{C}}^{\mathcal{Z}, \mathcal{I}}(\tilde{z} \mid z, I)$ can be interpreted as a stochastic map $\mathcal{Z} \to \widetilde{\mathcal{Z}}$. The above relation can then be written as a commuting diagram of stochastic maps, $\forall I \in \mathcal{I}, I' = \psi(I)$:

$$
\begin{array}{ccc}
\mathcal{Z} & \xrightarrow{p_{\mathcal{C}}^{\mathcal{Z}, \mathcal{I}}(\tilde{z}|z,I)} & \widetilde{\mathcal{Z}} \\
\downarrow{\varphi} & & \downarrow{\varphi} \\
\mathcal{Z}' & \xrightarrow{p_{\mathcal{C}'}^{\mathcal{Z}', \mathcal{I}'}(\tilde{z}'|z',I')} & \widetilde{\mathcal{Z}}'
\end{array}
\tag{10}
$$

where we treat $\varphi : \mathcal{Z} \to \mathcal{Z}'$ as a deterministic stochastic map.

For any variable $i \in [n]$, write the other nodes as $o = [n] \setminus \{i\}$. Let $I = \{i\}$. Then $p_{\mathcal{C}}^{\mathcal{Z}, \mathcal{I}}(\tilde{z} \mid z, I)$ can be written as a string diagram of stochastic maps:

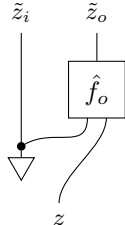

This string diagram represents a conditional probability distribution $p(\tilde{z}_i, \tilde{z}_o \mid z)$ and is read from the bottom to the top. String diagrams map formally to a generative process [26] and have been used previously in the context of causal models [27]. In this case, the diagram maps to:

$$\tilde{z}_i \sim p(\tilde{z}_i), \quad \tilde{z}_o = \hat{f}_o(\tilde{z}_i, z)$$

where $p(\tilde{z}_i)$ is the interventional distribution and the deterministic map $\hat{f}_o : \widetilde{\mathcal{Z}}_i \times \mathcal{Z} \to \widetilde{\mathcal{Z}}_o$ can be constructed from the inverse solution $s^{-1} : \mathcal{Z} \to \mathcal{E}$ and the causal mechanisms. Each box in a string diagram of stochastic maps denotes a stochastic map and each line to a measurable space. The triangle is the stochastic map $\star \to \widetilde{\mathcal{Z}}_i$ (the star denoting the one-point space; maps from which correspond to probability distributions over the codomain). The $\bullet$ represents copying a variable.

The above commuting diagram (10) can then be written as the equality of the following two string diagrams, where $\psi(I) = I' = \{i'\}, o' = [n] \setminus \{i'\}$. We write $\varphi : \mathcal{Z} \to \mathcal{Z}'$ as the pair $\varphi_{i'} : \mathcal{Z} \to \widetilde{\mathcal{Z}}'_{i'}, \varphi_{o'} : \mathcal{Z} \to \mathcal{Z}'_{o'}$ obtained by projecting the output of $\varphi$ to the partition $\mathcal{Z}' = \mathcal{Z}'_{i'} \times \mathcal{Z}'_{o'}$:

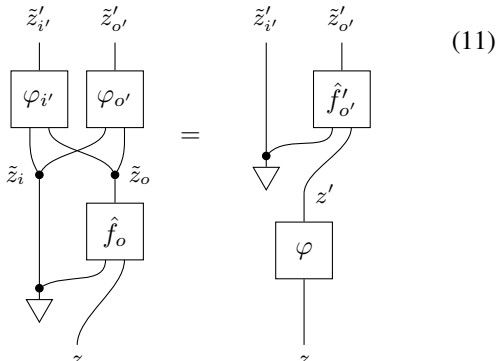

$$(11)$$

This should be read as the equality of the two conditional probability distributions $p(\tilde{z}'_{i'}, \tilde{z}'_{o'} \mid z)$ generated in the following way:

Left: $\quad \tilde{z}_i \sim p(\tilde{z}_i), \quad \tilde{z}_o = \hat{f}_o(\tilde{z}_i, z), \quad \tilde{z}'_{i'} = \varphi(\tilde{z}_i, \tilde{z}_o)_{i'}, \quad \tilde{z}'_{o'} = \varphi(z)_{o'}.$

Right: $\quad z' = \varphi(z), \quad \tilde{z}'_{i'} \sim p'(\tilde{z}'_{i'}), \quad \tilde{z}'_{o'} = f'_{o'}(\tilde{z}'_{i'}, z').$

The string diagram equality (11) implies equality when we disregard outputs $\mathcal{Z}'_{o'}$:

where the upwards pointing triangle represents discarding a variable.

Using Lemma 2, and the fact that $\widetilde{\mathcal{Z}}_i = \widetilde{\mathcal{Z}}'_{i'} = \mathbb{R}$, the composed differentiable function $\widetilde{\mathcal{Z}}_i \times \mathcal{Z} \to \widetilde{\mathcal{Z}}'_{i'}$ is constant in $\mathcal{Z}$. Thus we have a deterministic function $\widetilde{\mathcal{Z}}_i \to \widetilde{\mathcal{Z}}'_{i'}$ such

that:

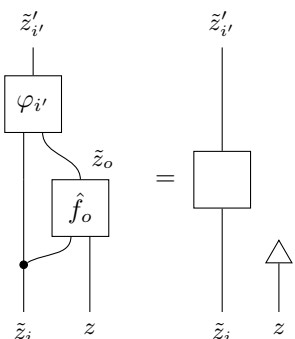

The deterministic function $\widetilde{\mathcal{Z}}_i \times \mathcal{Z} \to \widetilde{\mathcal{Z}}_i \times \widetilde{\mathcal{Z}}_o$ is surjective and both the left- and right-hand side can be seen as first applying this function (though the output is discarded on the right hand side), which implies there exists a function $\widetilde{\mathcal{Z}}_i \to \widetilde{\mathcal{Z}}'_{i'}$ such that

In words, the function $\varphi_{i'} : \mathcal{Z}_i \times \mathcal{Z}_o \to \mathcal{Z}'_{i'}$ is constant in $\mathcal{Z}_o$. This holds for all $i$ and thus $\varphi$ is $\psi$-diagonal.

**Step 3: Ancestry preserved** Let $i \neq j \in [n]$, $i' = \psi(i)$, $j' = \psi(j)$, and $I = \{i\}$. Writing $\varphi$ as $\psi$-diagonal, the commuting diagram (10) for the $j'$ component of $\tilde{z}'$, can be written as the following string diagram:

The left hand side is a deterministic map $\mathcal{Z} \to \widetilde{\mathcal{Z}}'_{j'}$ if and only if $\hat{f}_j$ is constant in $\widetilde{\mathcal{Z}}_i$ which by faithfulness is the case if and only if $i \notin \mathbf{anc}_j$. The same holds on the right hand side, so $\forall i \neq j \in [n], i \in \mathbf{anc}^{\mathcal{C}}_j \iff \psi(i) \in \mathbf{anc}^{\mathcal{C}'}_{\psi(j)}$.

**Step 4: Noise map diagonal** Define $\varphi_{\mathcal{E}} = s'^{-1} \circ \varphi \circ s : \mathcal{E} \to \mathcal{E}'$. Note that $\varphi_{\mathcal{E}}(\epsilon)_{i'}$ only depends on $\epsilon_i$ and $\epsilon_{\mathbf{anc}_i}$, because $s(\epsilon)_{\mathbf{anc}_i, i}$ and $s'^{-1}(z')_{i'}$ only depend on ancestors, $\varphi$ is $\psi$-diagonal and $\psi$ preserves ancestry.

The map $\varphi$ is measure-preserving. Thus $\forall i$ and writing $A = \mathbf{anc}_i$, the conditional $p(z_i \mid z_A) = p(z_i \mid z_{\mathbf{pa}_i})$, interpreted as a stochastic map, is preserved by $\varphi$. We can express this as another commuting diagram, in which the two paths from $\mathcal{E}_A$ to $\mathcal{E}'_{i'}$ must be equal:

$$
\begin{array}{ccccc}
\mathcal{E}_A & \xrightarrow{s_A} & \mathcal{Z}_A & \xrightarrow{p(z_i|z_{\mathbf{pa}_i})} & \mathcal{Z}_{A,i} \\
& & \downarrow{\varphi_A} & & \downarrow{\varphi_{A,i}} \\
& & \mathcal{Z}'_{A'} & \xrightarrow[p(z'_{i'}|z'_{\mathbf{pa}_{i'}})]{} & \mathcal{Z}'_{A',i'} \xrightarrow{f'^{-1}_{i'}} \mathcal{E}'_{i'}
\end{array}
$$

where $f'^{-1}_{i'}(z') = f(z'_{\mathbf{pa}_{i'}}, \cdot)^{-1}(z'_{i'})$. Then we have:

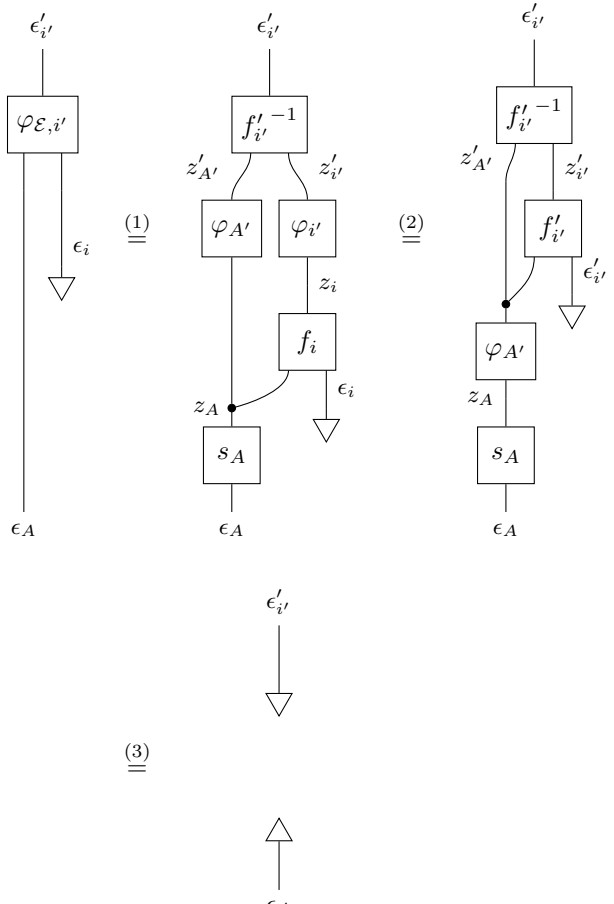

where the first equality follows from the definition of $\varphi_{\mathcal{E},i'}$, the second equality from the commuting diagram above and the third equality from the fact that $f'_{i'}$ and $f'^{-1}_{i'}$ cancel. Then, again using Lemma 2, the map on the left hand side must be constant in $\epsilon_A$. The noise encoding is thus also $\psi$-diagonal.

**Step 5: Equivalence** Consider for a variable $i$ and with $i' = \psi(i)$ the following commuting diagram of deterministic maps. Note that we write the causal mechanism $f_i$ as a

function of all ancestors, not just the parents, so it is constant in the non-parents. Because of faithfulness, it is non-constant in the parents. Since $\psi$ preserves ancestors, $f'_{i'}$ is well-typed.

$$
\begin{array}{ccc}
\mathcal{E} & \xrightarrow{\varphi_{\mathcal{E}}} & \mathcal{E}' \\
\downarrow{(s_{\mathbf{anc}_i}, \mathrm{id}_{\mathcal{E}_i})} & & \downarrow{(s'_{\mathbf{anc}_{i'}}, \mathrm{id}_{\mathcal{E}'_{i'}})} \\
\mathcal{Z}_{\mathbf{anc}_i} \times \mathcal{E}_i & \xrightarrow{(\varphi_{\mathcal{Z},\mathbf{anc}_i}, \varphi_{\mathcal{E},i})} & \mathcal{Z}'_{\mathbf{anc}_{i'}} \times \mathcal{E}'_{i'} \\
\downarrow{f_i} & & \downarrow{f'_{i'}} \\
\mathcal{Z}_i & \xrightarrow{\varphi_{\mathcal{Z},i}} & \mathcal{Z}'_{i'}
\end{array}
$$

The composition of the left vertical maps is equal to $s_i$, the composition of the right vertical maps to $s'_{i'}$. Therefore and because of the definition of $\varphi_{\mathcal{E}}$, the outer and the top square commute. Then, because $(s_{\mathbf{anc}_i}, \mathrm{id}_{\mathcal{E}_i})$ is surjective, the bottom square also commutes [28, Lemma 1.6.21].

Then for $z_j \in \mathbf{anc}_i$, we have that

$$
\begin{aligned}
z_j \in \mathbf{pa}_i^{\mathcal{C}} &\iff f_i \text{ not constant in } z_j \\
&\iff f'_{i'} \text{ not constant in } z'_{j'} \\
&\iff z'_{j'} \in \mathbf{pa}_{i'}^{\mathcal{C}'}.
\end{aligned}
$$

And thus $\psi$ not only preserves ancestry, but also parenthood and is thus a graph isomorphism $\psi : \mathcal{G}(\mathcal{C}) \to \mathcal{G}(\mathcal{C}')$. Diagram (5) commutes, and we have established an SCM isomorphism $\varphi : \mathcal{C} \to \mathcal{C}'$.

To have this also be an ISCM isomorphism, we need diagram (6) to commute and the distribution over interventions to be preserved. For the first, use the fact that all maps in (6) are isomorphisms to simply define $\tilde{\varphi}_{\mathcal{E}}$ so that the diagram commutes. The second follows directly from the assumptions. Hence $\varphi : \mathcal{D} \to \mathcal{D}'$ is an ISCM isomorphism, $\mathcal{D} \sim \mathcal{D}'$, and—together with the arguments in the beginning of this proof—finally $\mathcal{M} \sim \mathcal{M}'$. $\qquad\square$

## B LIMITATIONS & GENERALIZATION

Our identifiability result relies on a few assumptions. Here we discuss some key requirements of Thm. 1 and whether they can be relaxed.

**Diffeomorphic causal mechanisms** In Def. 4, we require causal mechanisms to be pointwise diffeomorphisms from noise variables to causal variables. Under some mild smoothness assumptions, any SCM can be brought into this form by elementwise redefinitions of the variables, without affecting the observational or interventional distributions. However, such a redefinition may change counterfactual / weakly supervised distributions.

**All interventions observed** To guarantee identifiability, we require that the intervention distribution has support for any atomic intervention: datasets need to contain data pairs generated from interventions on any causal variable. However, in many systems not all variables can be intervened upon, for instance because some variables are fundamentally immutable or for safety reasons. In that case, the LCM may not be fully identifiable, but there may still be partial identifiability. Interventions on child variables, for instance, can guarantee the identifiability of the parents [14]. We leave a precise characterization of the equivalence classes under such partial weak supervision for future work.

**Perfect interventions** Our proof of Thm. 1 requires perfect interventions, i.e. that intervened-upon mechanisms do not depend on any causal variables. This is arguably the biggest mismatch between our assumptions and many real-world systems.

If we try to generalize the assumptions and allow the intervention mechanisms to depend on the parent variables, identifiability is lost. A simple counterexample is the following.

**Example 1** (Non-identifiable ISMCs under imperfect interventions). *Consider two inequivalent ISCMs $\mathcal{D}, \mathcal{D}'$, each with two variables and graph $z_1 \rightarrow z_2$. Let the mechanisms be equal, except that $f_2'(\epsilon_2; z_1) = f_2(\epsilon_2; z_1) + z_1$ and $\tilde{f}_2'(\tilde{\epsilon}_2; z_1) = \tilde{f}_2(\tilde{\epsilon}_2; z_1) + z_1$. Then it is easy to see that a $\phi$ that is the identity on the first variable and on the second variable is $\phi_2(z_1, z_2) = z_1 + z_2$, preserves the weakly supervised distribution, but is not an ISCM isomorphism.*

**Diffeomorphic decoder** Definition 1 and Thm. 1 assume that the map from causal variables to observed data is given by a deterministic, diffeomorphic decoder. However, our practical implementation in a VAE uses a stochastic decoder and allows for noisy data. Our experiments provide empirical evidence for identifiability in this setting. We believe that it may be possible to extend Thm. 1 to stochastic decoders, similarly to Khemakhem et al. [8]. We plan to study this extension in future work.

**Real-valued causal variables** Theorem 1 assumes real-valued causal and noise variables, $\mathcal{Z}_i = \mathcal{E}_i = \mathbb{R}$. We can easily extend this to intervals $(a, b) \in \mathbb{R}$, as these are isomorphic to $\mathbb{R}$. However, the extension to arbitrary continuous spaces or $\mathbb{R}^n$ is not straightforward. The main reason is that our proof relies on Lemma 2, which does not generalize.

Let us provide a counterexample for identifiability with circle $S^1$-valued causal variables.

**Example 2** ($S^1$-valued non-identifiable LCMs). *Consider an LCM $\mathcal{M} = \langle \mathcal{C}, \mathcal{X}, g, \mathcal{I}, p_{\mathcal{I}} \rangle$ with the following components:*

- *The SCM $\mathcal{C}$ consists of two circle-valued variables $z_1, z_2 \in S^1$ with noise variables $\epsilon_1, \epsilon_2 \in S^1$. We pa-*

rameterize $S^1$ as $[0, 2\pi)$ with addition defined modulo $2\pi$.

- *The causal graph is $z_1 \rightarrow z_2$.*
- *The causal mechanisms are $f_1(\epsilon_1) = \epsilon_1$ and $f_2(\epsilon_2; z_1) = \epsilon_2 + z_1$.*
- *The solution function is $s(\epsilon_1, \epsilon_2) = (\epsilon_1, \epsilon_2 + \epsilon_1)$.*
- *The noise variables are distributed as $\epsilon_1 \sim \mathcal{U}$, uniformly, and $\epsilon_2 \sim q$, which we require to not be invariant under translations (so in particular not uniform). For example, one can take the von Mises distribution $\log q(\epsilon_2) = \cos(\epsilon_2) + \text{const}$.*
- *The observation space is $\mathcal{X}$ and the decoder $g : S^1 \times S^1 \rightarrow \mathcal{X}$ is diffeomorphic.*
- *The intervention set $\mathcal{I}$ consists of the empty intervention, atomic interventions on $z_1$ with $\tilde{z}_1 \sim \mathcal{U}$, and atomic interventions on $z_2$ with $\tilde{z}_2 \sim \mathcal{U}$. Each of these interventions has probability $\frac{1}{3}$ in $p_{\mathcal{I}}$.*

*Note that the SCM is faithful, as $z_1 \not\perp\!\!\!\perp z_2$ in the observational distribution, because $q$ is not translationally invariant. The LCM entails the weakly supervised causal distribution*

$$
p_{\mathcal{C}}^{\mathcal{Z}}(z, \tilde{z}) = \mathcal{U}(z_1) \, q(z_2 - z_1) \left[ \frac{1}{3} \, \delta(\tilde{z}_1 - z_1) \, \delta(\tilde{z}_2 - z_2) \right.
$$
$$
\left. + \frac{1}{3} \, \mathcal{U}(\tilde{z}_1) \, \delta(\tilde{z}_2 - z_2 - \tilde{z}_1 + z_1) + \frac{1}{3} \, \delta(\tilde{z}_1 - z_1) \, \mathcal{U}(\tilde{z}_2) \right]
$$

(12)

*with Dirac delta $\delta$. The weakly supervised data distribution is then given by $p_{\mathcal{M}}^{\mathcal{X}} = (g_*, g_*) p_{\mathcal{C}}^{\mathcal{Z}}$.*

*Now consider a second LCM $\mathcal{M}' = \langle \mathcal{C}', \mathcal{X}, g', \mathcal{I}', p_{\mathcal{I}'}' \rangle$:*

- *The SCM $\mathcal{C}'$ consists of two circle-valued variables $z_1', z_2' \in S^1$ with noise variables $\epsilon_1', \epsilon_2' \in S^1$.*
- *The causal graph is trivial and the causal mechanisms are given by the identity, $f_i'(\epsilon_i') = \epsilon_i'$.*
- *The noise variables are distributed as $\epsilon_1' \sim \mathcal{U}$ and $\epsilon_2' \sim q$.*
- *The observation space is $\mathcal{X}$ and the decoder $g' : S^1 \times S^1 \rightarrow \mathcal{X}$ is given by the diffeomorphism $g'(z') = g \circ s(z')$, where $s$ is the solution function of $\mathcal{C}$.*
- *The intervention set $\mathcal{I}'$ consists of empty interventions, atomic interventions on $z_1'$ with $\tilde{z}_1' \sim \mathcal{U}$, and atomic interventions on $z_2'$ with $\tilde{z}_2' \sim \mathcal{U}$. Each of these interventions has probability $\frac{1}{3}$ in $p_{\mathcal{I}'}$.*

*We find a weakly supervised causal distribution*

$$
p_{\mathcal{C}'}^{\mathcal{Z}'}(z', \tilde{z}') = \mathcal{U}(z_1') \, q(z_2') \left[ \frac{1}{3} \, \delta(\tilde{z}_1' - z_1') \, \delta(\tilde{z}_2' - z_2') \right.
$$
$$
\left. + \frac{1}{3} \, \mathcal{U}(\tilde{z}_1') \, \delta(\tilde{z}_2' - z_2') + \frac{1}{3} \, \delta(\tilde{z}_1' - z_1') \, \mathcal{U}(\tilde{z}_2') \right]. \quad (13)
$$

*Clearly, two LCMs are not equivalent, because their graphs are non-isomorphic. Yet, if we define*

$$\varphi : \mathcal{Z} \to \mathcal{Z}' : (z_1, z_2) \mapsto (z_1, z_2 - z_1)$$

*then the weakly supervised distribution of the causal variables is preserved:*

$$
\begin{aligned}
&((\varphi, \varphi)_* p_{\mathcal{C}}^{\mathcal{Z}})(z', \tilde{z}') \\
&= p_{\mathcal{C}}^{\mathcal{Z}}((z_1', z_2' + z_1'), (\tilde{z}_1', \tilde{z}_2' + \tilde{z}_1')) \\
&= \mathcal{U}(z_1') \, q(z_2' + z_1' - z_1') \\
&\quad \times \Bigg[ \frac{1}{3} \, \delta(\tilde{z}_1' - z_1') \, \delta(\tilde{z}_2' + \tilde{z}_1' - (z_2' + z_1')) \\
&\qquad + \frac{1}{3} \, \mathcal{U}(\tilde{z}_1') \, \delta(\tilde{z}_2' + \tilde{z}_1' - (z_2' + z_1') - \tilde{z}_1' + z_1') \\
&\qquad + \frac{1}{3} \, \delta(\tilde{z}_1' - z_1') \, \mathcal{U}(\tilde{z}_2' + \tilde{z}_1') \Bigg] \\
&= \mathcal{U}(z_1') \, q(z_2') \\
&\quad \times \Bigg[ \frac{1}{3} \, \delta(\tilde{z}_1' - z_1') \, \delta(\tilde{z}_2' - z_2') \\
&\qquad + \frac{1}{3} \, \mathcal{U}(\tilde{z}_1') \, \delta(\tilde{z}_2' - z_2') \\
&\qquad + \frac{1}{3} \, \delta(\tilde{z}_1' - z_1') \, \mathcal{U}(\tilde{z}_2') \Bigg] \\
&= p_{\mathcal{C}'}^{\mathcal{Z}'}(z', \tilde{z}')
\end{aligned}
$$

*where we use that the density $\mathcal{U}$ is constant. Also, because $\varphi = s^{-1}$ and $g'(z') = g \circ s(z')$, we have that $p_{\mathcal{M}}^{\mathcal{X}} = p_{\mathcal{M}'}^{\mathcal{X}}$.*

*So these two models with their non-isomorphic graph structures have identical weakly-supervised distributions on the observables $x, \tilde{x}$. They therefore provide a counter-example for a straightforward generalization of Thm. 1 to causal variables with arbitrary continuous domains.*

The key issue here is that the interventional distribution on $\tilde{z}_2$ has many symmetries or automorphisms: diffeomorphic maps $\mathcal{Z}_2 \to \mathcal{Z}_2$ that preserve $p(\tilde{z}_2)$ — in this case these are the cyclic translations. In general, for any causal model $\mathcal{D}$ with two variables $z_1 \to z_2$, we can construct a map $\phi(z_1, z_2) = (z_1, \Gamma(z_1)(z_2))$ where for all $z_1$, $\Gamma(z_1) : \mathcal{Z}_2 \to \mathcal{Z}_2$ is a differentiable map from $\mathcal{Z}_1$ to a diffeomorphism on $\mathcal{Z}_2$ that preserves the interventional distribution $p(\tilde{z}_2)$. This map $\phi$ preserves the weakly supervised distribution from $\mathcal{D}$ to a unique model $\mathcal{D}'$, whose causal mechanisms are: $f_1'(\epsilon_1) = f_1(\epsilon_1)$ and $f_2'(z_1, \epsilon_1) = \Gamma(z_1)(f_2(z_1, \epsilon_2))$. However, $\phi$ is only an ISCM morphism if it is also diagonal, and thus $\Gamma$ must be constant in $z_1$.

For the $\mathbb{R}$-valued variables of the main paper, any smooth distribution on $\mathbb{R}$ has exactly two automorphisms, related to the automorphisms of the univariate Gaussian distribution $x \mapsto x$ and $x \mapsto -x$. $\Gamma$ can not smoothly switch between these and thus must be constant, making $\phi$ diagonal and an ISCM morphism. However, any multi-dimensional distribu-

tion has many automorphisms. One class of these is related to the orthogonal transformations of a standard multivariate Gaussian. Another, much larger, class is related to the flows generated by divergence-free vector fields on the unit ball. $\Gamma$ can smoothly choose different such automorphisms for different values of $z_1$, making $\phi$ not diagonal and thus not an ISCM morphism. In conclusion, this smooth space of automorphisms make the multivariate case unidentifiable from weak supervision.

# C  IMPLICIT LATENT CAUSAL MODELS

## C.1  MODEL SPECIFICATION

An implicit latent causal model for a system of $n$ causal variables consists of the following components, see Fig. 6:

- a Gaussian noise encoder $q(e|x)$ with mean $\mu_e(x)$ and standard deviation $\sigma_e(x)$ implemented as neural networks;

- a Gaussian noise decoder $p(x|e)$ with mean $\mu_x(e)$ implemented as neural network and fixed, constant standard deviation;

- an intervention encoder $q(I|x, \tilde{x})$ defined as

$$
\begin{aligned}
&\log q(i \in I | x, \tilde{x}) = \\
&\frac{1}{Z} \left( a + b \, |\mu_e(x)_i - \mu_e(\tilde{x})_i| + c \, |\mu_e(x)_i - \mu_e(\tilde{x})_i|^2 \right),
\end{aligned}
$$

  where $a < 0$ and $b, c > 0$ are learnable parameters and where the normalization constant $Z$ is defined such that $\sum_I q(i \in I | x, \tilde{x}) = 1$;

- solution functions $s_i(e_i; e_{\setminus i})$ for $i = 1, \ldots, n$ implemented as invertible affine transformations, where the offset and slope are functions of $e_{\setminus i}$ implemented with neural networks;

- noise priors $p_i(e_i)$, which we choose to be standard Gaussian;

- an post-intervention causal-variable prior $\tilde{\pi}(\tilde{z}_i)$, which we choose to be standard Gaussian; and

- an intervention-target prior $p(I)$, which we choose to be uniform.

Encoding a data pair $(x, \tilde{x})$ during training consists of the

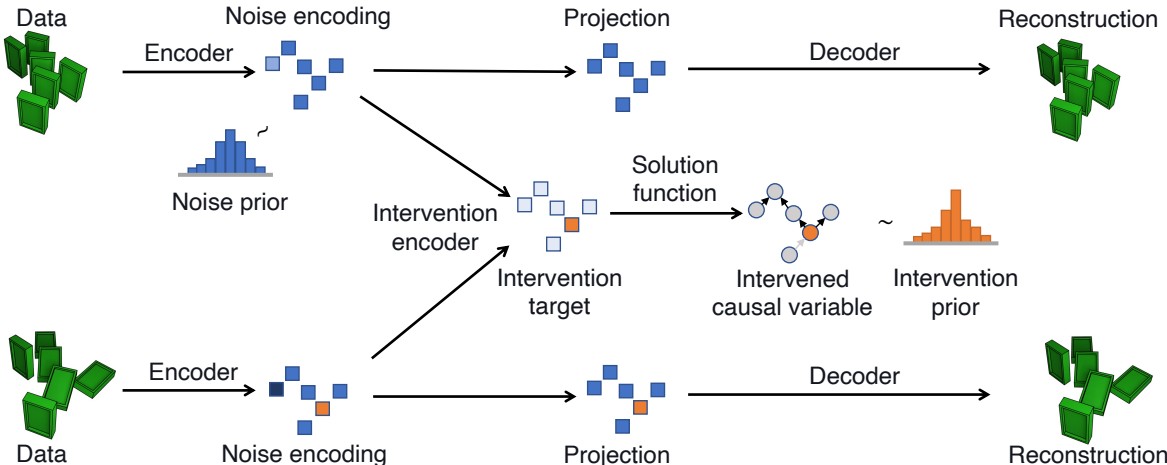

Figure 6: ILCM architecture. Pre- and post-intervention data (left) are encoded to noise encodings and intervention targets, which are then decoded back to the data space. To compute the prior probability density, the noise encodings are transformed into causal variables with the neural solution function.

following steps:

$$I \sim q(I|x, \tilde{x})$$
$$e_{\text{preliminary}} \sim q(e|x)$$
$$\tilde{e}_{\text{preliminary}} \sim q(\tilde{e}_{\text{preliminary}}|\tilde{x})$$
$$\forall i, \ \lambda_i \sim \text{Uniform}(0, 1)$$
$$e_{\text{average } i} = \lambda_i e_{\text{preliminary } i} + (1 - \lambda_i)\tilde{e}_{\text{preliminary } i}$$
$$e_i = \begin{cases} e_{\text{preliminary } i} & i \in I \\ e_{\text{average } i} & i \notin I \end{cases}$$
$$\tilde{e}_i = \begin{cases} \tilde{e}_{\text{preliminary } i} & i \in I \\ e_{\text{average } i} & i \notin I \end{cases}. \tag{14}$$

In the first line, we encode to an intervention target (either by sampling or by enumerating all possibilities and summing up the corresponding loss terms). In the second and third line, the data is encoded to the noise-encoding space. The fourth to last lines project these noise encodings such that for those components $e_i$ that are not intervened upon, $i \notin I$, we have that the pre-intervention noise encoding and post-intervention noise encoding are equal, $e_i = \tilde{e}_i$. This makes sure that the latents are consistent with the weakly supervised structure, and punishes deviations from this structure through the reconstruction error (likelihood). We use the symbol $e, \tilde{e} \sim q(e, \tilde{e}|x, \tilde{x}, I)$ to refer to the fourth to last line together.

The prior is given by

$$p(e, \tilde{e}_I, I) = p(I) \prod_i p_i(e_i) \prod_{i \in I} p(\tilde{e}_i|e)$$

with, for $i \in I$,

$$p(\tilde{e}_i|e) = \tilde{p}\big(s_i(\tilde{e}_i|e_{\setminus i})\big) \left| \frac{\partial \bar{s}_i(\tilde{e}_i; e_{\setminus i})}{\partial \tilde{e}_i} \right|. \tag{15}$$

## C.2 TRAINING

**Loss**  We train ILCMs by minimizing the $\beta$-VAE loss

$$\mathcal{L}_{\text{ILCM}} = \mathbb{E}_{x,\tilde{x}} \mathbb{E}_{I \sim q(I|x,\tilde{x})} \mathbb{E}_{e,\tilde{e} \sim q(e,\tilde{e}|x,\tilde{x},I)}$$
$$\left[ \log p(x|e) + \log p(\tilde{x}|\tilde{e}) \right.$$
$$\left. + \beta \Big\{ \log p(e, \tilde{e}_I, I) - \log q(I|x,\tilde{x}) - \log q(e, \tilde{e}_I|x, \tilde{x}, I) \Big\} \right],$$

where $\beta$ is a hyperparameter.

Two additional loss terms are used as regularizers during training. The first is a plain reconstruction error (or log likelihood term) that does not use the projections given in Eq. (14) to encourage consistency between the noise encoder and noise decoder:

$$\mathcal{L}_{\text{reco}} = \mathbb{E}_{x,\tilde{x}} \mathbb{E}_{e \sim q(e|x)} \mathbb{E}_{\tilde{e} \sim q(\tilde{e}|\tilde{x})} \left[ \log p(x|e) + \log p(\tilde{x}|\tilde{e}) \right].$$

Throughout training, we also add the negative entropy of the batch-aggregate intervention posterior $q_I^{\text{batch}}(I) = \mathbb{E}_{x,\tilde{x} \in \text{batch}}[q(I|x,\tilde{x})]$:

$$\mathcal{L}_{\text{entropy}} = \mathbb{E}_{\text{batches}} \left[ -\sum_I q_I^{\text{batch}}(I) \log q_I^{\text{batch}}(I) \right].$$

This helps avoid a collapse of the latent space to a lower-dimensional subspace. The overall loss is then given by $\mathcal{L} = \mathcal{L}_{\text{ILCM}} + \alpha \mathcal{L}_{\text{reco}} + \gamma \mathcal{L}_{\text{entropy}}$ with hyperparameters $\alpha, \gamma \geq 0$.

**Training phases**  We train ILCM models in four phases:

1. We begin with a short pre-training phase, in which the noise encoder and noise decoder are trained on a plain $\beta$-VAE loss with a standard Gaussian prior. This provides a good starting point for the remainder of

the training, in which the encoders and decoders have already learned some patterns in the data space.

2. Next, we train the noise encoder, noise decoder, and intervention encoder on the combined loss $\mathcal{L}$ described above. However, in this phase we do not yet use the solution functions and instead model $p(\tilde{e}_i|e)$ with a uniform probability density. This avoids a feedback the randomly initialized solution functions influencing the training of the encoder and decoders, stabilizing the learning of good latent representations.

3. We then "switch on" the solution functions and model the intervention targets with the density $p(\tilde{e}_i|e)$ as given above and train on the combined loss $\mathcal{L}$.

   On image datasets, we freeze the convolutional layers in the encoder in this stage and only continue training the final layers of the encoder together with the solution functions.

4. For the final training phase, we change the setup in two more ways. First, we analyze the learned solution functions to infer the most likely topological order. For this we use the step 1 of the heuristic algorithm for graph inference, which we will define in Sec. C.4 below. Then the solution functions are modified such that $s_i$ only depends on the ancestors $e_{\mathbf{anc}_i}$ according to the inferred topological order:

$$s_i(e_i; e_{\setminus i}) \rightarrow s_i(e_i; e_{\mathbf{anc}_i}) . \tag{16}$$

   This is implemented with a suitable masking layer in the neural network implementation of $s_i$. We find that this form of inductive bias in the prior helps with learning cleanly disentangled representations.

   Second, we fix the intervention encoders to the deterministic,

$$q^{\text{deterministic}}(I|x,\tilde{x}) = \begin{cases} 1 & I = \arg\max_I q(I|x,\tilde{x}) \\ 0 & \text{else} . \end{cases}$$

   This further improves the training efficiency.

The separation of phase 2 and 3—first training the encoder with a simplified prior, then training the solution functions—improved the success of our method substantially. Adding the pre-training and fine-tuning phases 1 and 4 slightly improved the efficiency of the training, but is not critical.

We use the Adam optimizer [29] with a cosine annealing schedule [30], which is restarted at the beginning of training phases 3 and 4. The hyperparameters differ slightly between experiments and will be given in Sec. D.

## C.3 IDENTIFIABILITY

We will now show that our identifiability result extends to implicit latent causal models. First, note that any implicit

LCM, whose intervention distributions $p(\tilde{e}_i \mid e)$ satisfy some topological ordering so that $s$ is a diffeomorphism, corresponds exactly to one explicit LCM, by pre-composing the decoder with the solution diffeomorphism. The weakly supervised distribution of an ICLM and its equivalent ELCM are the same. Similarly, any ELCM isomorphism corresponds exactly to an appropriately defined morphism of ILCMs.

Using the language of category theory, we can visualize the relationship between explicit and implicit latent causal models in the following commuting diagram:

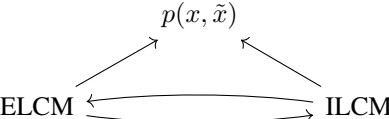

where ELCM is the category of explicit latent causal models and morphisms thereof, ILCM the isomorphic category of implicit latent causal models and its morphisms, and $p(x, \tilde{x})$ the discrete category whose objects are weakly supervised distributions and which contains only identity morphisms. We know that isomorphic ELCMs have the same weakly supervised distribution, so that we indeed get a functor ELCM $\rightarrow p(x, \tilde{x})$. Our main result furthermore says that this functor is full, so that any equality of weakly-supervised distributions comes from an isomorphism of ELCMs.

Because the categories of ELCMs and ICLMs are isomorphic, this result, and thus identifiability, also holds for the equivalent ILCM formulation.

## C.4 GRAPH INFERENCE

ILCMs do not contain an explicit graph representation, but the causal structure is implicitly represented by the learned solution functions. We propose two algorithms for causal discovery based on a trained ILCM model.

**ILCM-E** We can perform causal discovery in a two-stage procedure. After training an ILCM to learn the causal representations, we use an off-the-shelf method for causal discovery on the learned representations. Since the ILCM allows us to infer intervention targets, we can use intervention-based algorithms. In this paper, we use ENCO [18], a recent differentiable causal discovery method that exploits interventions to obtain acyclic graphs without requiring constrained optimization. Alternatives to ENCO include DCDI [19] and GIES [20].

**ILCM-H** Alternatively, we can unearth the causal structure encoded in the learned solution functions $s_i$, which map noise variables to causal variables. We introduce a heuristic algorithm that can find the causal graph $\mathcal{G}$ and the causal mechanisms $f_i$ from a trained ILCM. It proceeds in three steps:

1. *Computing the topological order*: To determine the topological order of the causal graph, we use the property that after convergence, the solution function $z_i = s_i(e_i; e)$ will only depend on the ancestors of $e_i$. This allows us to define a heuristic that determines whether $z_i$ is an ancestor of $z_j$. For each pair $(i, j)$, we compute

$$\text{ancestry}(i, j) = d\Big(s_j(e_j; e), s_j(e_j; \text{mask}_i[e])\Big). \tag{17}$$

Here $d(\cdot, \cdot)$ is a distance measure between two functions; in practice, we use the expected MSE over a validation dataset. The function $\text{mask}_i[e]$ replaces the $i$-th component of $e$ with uninformative input, for instance median of that component computed over the training dataset.

After computing these ancestry scores, we can compute a topological order by sorting the variables such that likely ancestors appear before their likely descendants according to this heuristic. We do this with a greedy algorithm.

2. *Extracting the causal mechanisms*: Next, we compute the causal mechanisms $f_i$ such that $z_i = f_i(e_i; z_{\mathbf{anc}_i})$. We begin by setting $i$ to the root node (the first variable in the topological order computed in step 1) and proceed in topological order. At every step we set

$$f_i(e_i; z_{\mathbf{anc}_i}) = s_i(e_i; \hat{e}) \tag{18}$$

with

$$\hat{e}_j = \begin{cases} f_j^{-1}(z_j; z_{\mathbf{anc}_j}) & j \in \mathbf{anc}_i \\ \text{mask}[e_j] & \text{otherwise}. \end{cases} \tag{19}$$

3. *Finding causal parents*: Finally, we check whether an ancestor $z_i$ is a parent of a node $z_j$ by testing whether $f_j$ explicitly depends on $z_i$. Again, we use a heuristic measure of functional dependence:

$$\text{paternity}(i, j) = d\Big(f_j(e_j; z_{\mathbf{anc}_j}), f_j(e_j; \text{mask}_i[z_{\mathbf{anc}_j}])\Big) \tag{20}$$

We then construct the causal graph by thresholding on this heuristic. This gives us the inferred adjacency matrix

$$\hat{A}_{ij} = \begin{cases} 1 & i \in \mathbf{anc}_j \quad \text{and} \quad \text{paternity}(i, j) > p_{\min} \\ 0 & \text{else} \end{cases} \tag{21}$$

where $p_{\min} > 0$ is a hyperparameter.

The heuristic algorithm (ILCM-H) does not require any optimization and is thus computationally more efficient, but arguably less principled than the likelihood-based ENCO approach (ILCM-E). In our experiments, the ILCM-H approach finds the correct graph in 7 out of the 8 datasets, while ILCM-E always yields the correct causal graph.

## D EXPERIMENTS

### D.1 GENERAL SETUP

**Baselines** We compare our ILCM-E and ILCM-H results (which differ only by the graph inference algorithm, as described above) to three baseline methods. The *disentanglement VAE* (dVAE) method is a VAE for paired data with individual latent components changing between the pre-intervention latents and the post-intervention latents, but without causal structure. We implement it by using an ILCM, but enforce a trivial causal graph by not allowing the solution functions $s_i(\cdot; e)$ to depend on $e$.

Our second baseline is an unstructured $\beta$-VAE, which treats pre-intervention and post-intervention data as i.i.d. and models both with a standard Gaussian prior.

Finally, we include a slot attention baseline. We use as many slots as there are latents. We break the symmetry between the slots by initialising the slots not with a random vector, but with a different learned vector per slot, as is done in Ref. [31]. We choose a six-dimensional latent for each slot.

**Metrics** We evaluate the disentanglement of the learned causal variables by computing the DCI disentanglement score [32]. There are many other disentanglement metrics, but empirically these tend to be highly correlated with the DCI disentanglement score [3], so we omit them here for simplicity. For the slot attention models, we add up the contribution of the latent dimensions of each slot, to get a importance matrix between slots and ground truth causal variables.

The quality of intervention inference is evaluated with the accuracy of the intervention encoder. Since we can only identify causal variables and intervention targets up to a permutation, we compute this accuracy for any possible permutation of the causal variables and then report the best result.

Finally, we evaluate the quality of the inferred causal graphs. We identify the ground-truth variables with the corresponding learned causal variables based on the importance matrix computed for the DCI disentanglement score [32]. We then compute the structural Hamming distance (or graph edit distance) between the learned graph and the true graph. As an example, consider the case of two causal variables, where the ground-truth graph is $z_1 \rightarrow z_2$, the ILCM graph is $z_1' \rightarrow z_2'$, and the ground-truth and learned variables are mapped to each other as $z_1 \leftrightarrow z_2'$ and $z_2 \leftrightarrow z_1'$. Then the structural Hamming distance will be 1, as the cause and effect are flipped in the learned model.

Table 2: Detailed experiment results. We compare our ILCM-E (using ENCO for graph inference) and ILCM-H (with a heuristic for graph inference) to disentanglement VAE (dVAE-E), unstructured $\beta$-VAE, and slot attention baselines. We show the DCI disentanglement score, the accuracy of intervention inference, the learned graph, and the structural Hamming distance (SHD) between learned and true graph. Best results in bold.

| Dataset | True graph | Method | Disentanglement | Intervention accuracy | Learned graph | SHD |
|---|---|---|---|---|---|---|
| 2D toy data | ○→○ | ILCM-E (ours) | **0.99** | **0.96** | ○→○ | **0** |
| | | ILCM-H (ours) | **0.99** | **0.96** | ○→○ | **0** |
| | | dVAE | 0.35 | **0.96** | ○ ○ | 1 |
| | | $\beta$-VAE | 0.52 | – | n/a | – |
| Causal3DIdent | ○ ○ ○ | ILCM-E (ours) | 0.99 | **0.98** | ○ ○ ○ | **0** |
| | | ILCM-H (ours) | 0.99 | **0.98** | ○ ○ ○ | **0** |
| | | dVAE | **1.00** | **0.98** | ○ ○ ○ | **0** |
| | | $\beta$-VAE | 0.94 | – | n/a | – |
| | | Slot attention | 0.90 | – | n/a | – |
| | ○→○ ○ | ILCM-E (ours) | **1.00** | **0.98** | ○→○ ○ | **0** |
| | | ILCM-H (ours) | **1.00** | **0.98** | ○→○ ○ | **0** |
| | | dVAE | 0.91 | **0.98** | ○ ○ ○ | 1 |
| | | $\beta$-VAE | 0.92 | – | n/a | – |
| | | Slot attention | 0.56 | – | n/a | – |
| | ○→○ ○ | ILCM-E (ours) | **0.99** | **0.98** | ○→○ ○ | **0** |
| | | ILCM-H (ours) | **0.99** | **0.98** | ○→○ ○ | **0** |
| | | dVAE | 0.83 | **0.98** | ○→○ ○ | 2 |
| | | $\beta$-VAE | 0.63 | – | n/a | – |
| | | Slot attention | 0.42 | – | n/a | – |
| | ○→○→○ | ILCM-E (ours) | **0.99** | **0.98** | ○→○→○ | **0** |
| | | ILCM-H (ours) | **0.99** | **0.98** | ○→○ ○ | 1 |
| | | dVAE | 0.79 | **0.98** | ○ ○ ○ | 2 |
| | | $\beta$-VAE | 0.63 | – | n/a | – |
| | | Slot attention | 0.86 | – | n/a | – |
| | ○ ○→○ | ILCM-E (ours) | **0.99** | **0.98** | ○ ○→○ | **0** |
| | | ILCM-H (ours) | **0.99** | **0.98** | ○ ○→○ | **0** |
| | | dVAE | 0.80 | **0.98** | ○ ○ ○ | 2 |
| | | $\beta$-VAE | 0.28 | – | n/a | – |
| | | Slot attention | 0.32 | – | n/a | – |
| | ○→○→○ | ILCM-E (ours) | **0.99** | **0.98** | ○→○→○ | **0** |
| | | ILCM-H (ours) | **0.99** | **0.98** | ○→○→○ | **0** |
| | | dVAE | 0.60 | **0.98** | ○ ○ ○ | 3 |
| | | $\beta$-VAE | 0.57 | – | n/a | – |
| | | Slot attention | 0.53 | – | n/a | – |
| CausalCircuit | ○→○→○○ | ILCM-E (ours) | **0.97** | **1.00** | ○→○→○○ | **0** |
| | | ILCM-H (ours) | **0.97** | **1.00** | ○→○→○○ | **0** |
| | | dVAE-E | 0.34 | **1.00** | ○ ○ ○ | 5 |
| | | $\beta$-VAE | 0.39 | – | n/a | – |
| | | Slot attention | 0.39 | – | n/a | – |

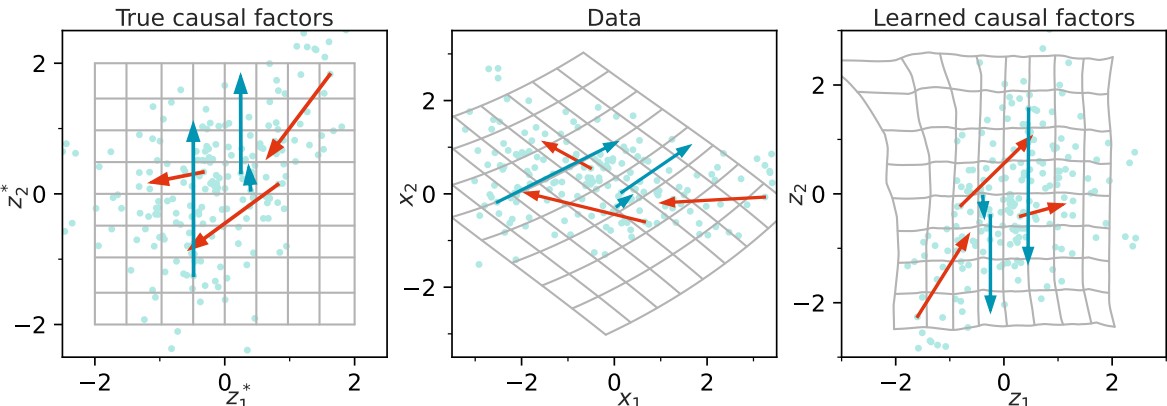

Figure 7: 2D toy data with graph $z_1^* \to z_2^*$. The grey grids show the map between true causal factors, data, and latent causal factors learned by the LCM. The mint dots indicate the observational data distribution, the arrows from $z$ to $\tilde{z}$ show interventions targeting $z_1^*$ (red) or $z_2^*$ (blue). The fact that axis-aligned lines in the true latent space are mapped to axis-aligned lines in the learned latent space implies that the disentanglement succeeded.

## D.2 2D TOY EXPERIMENT

**Dataset** We first demonstrate LCMs in a pedagogical toy experiment with $\mathcal{X} = \mathcal{Z} = \mathbb{R}^2$. Latent data is generated from a nonlinear SCM with graph $z_1 \to z_2$. In particular, we have that $z_1 \sim \mathcal{N}(z_1; 0, 1^2)$ and $z_2 \sim \mathcal{N}(z_1; 0.3z_1^2 + 0.6z_1, 0.8^2)$. This latent data is mapped to the data space $\mathcal{X} = \mathbb{R}^2$ with a randomly initialized coupling flow with five affine coupling layers interspersed with random permutations of the dimensions. For the weakly supervised setting we use a uniform intervention prior over $\{\emptyset, \{z_1\}, \{z_2\}\}$. We generate $10^5$ training samples, $10^4$ validation samples, and $10^4$ evaluation samples (where each sample is one pair $(x, \tilde{x})$ of pre- and post-intervention data).

**Architecture** The noise encoder and noise decoder are Gaussian, with mean and standard deviation computed by fully connected networks. The solution functions are implemented as affine transformations with slope and offset computed as a function of the pre-intervention noise encodings, also implemented with fully connected networks. For each MLP, we use two hidden layers with 100 units each and ReLU activations.

**Training** Models are trained using the procedure described in Sec. C.2. We train for $9 \cdot 10^4$ steps using a batch size of 100 and an initial learning rate of $10^{-3}$. The weights of the different loss terms and regularizers are as follows: $\beta$ is initially set to 0 and increased to its final value of 1 during training, $\alpha = 10^{-2}$, and $\gamma = 0$ throughout training. For each method, we train models with three random seeds and in the end select the median run according to the validation loss.

**Results** An ILCM trained in the weakly supervised setting is able to reconstruct the causal factors accurately up to elementwise reparameterizations, as shown in Fig. 7. In

Tbl. 2 we quantify the quality of the learned representations with the DCI disentanglement score [32]. We find that our LCM is able to disentangle the causal factors almost perfectly, while the baselines, which assume independent factors of variation, fail as expected. Both the ILCM and the dVAE baseline infer the intervention targets with high accuracy. Finally, we test the quality of the learned causal graphs. We infer the implicit graph with ENCO and the heuristic algorithm discussed above. In both cases, the learned causal graph is identical to the correct one, whereas the representations found by the dVAE baseline induce a wrong graph.

## D.3 CAUSAL3DIDENT EXPERIMENTS

**Dataset** In the Causal3DIdent experiments we consider six different datasets, each generated from a different causal graph, SCM, and decoder. The six causal graphs we consider are:

- the trivial graph ○ ○ ○,
- single edge ○↗○ ○,
- the chain ○↗○↘○,
- the fork ○↙○↘○,
- the collider ○↘○↙○, and
- the full graph ○↘○↙○.

For each of these subsets, we randomly generate a nonlinear SCM with heteroskedastic noise: for each causal mechanism, we randomly initialize an MLP that outputs the scale and shift of an affine transformation as a function of the causal parents. We choose an MLP initialization scheme that emphasizes nontrivial, nonlinear causal effects. We then identify a random permutation of the three causal variables with three high-level concepts in the Causal3DIdent dataset:

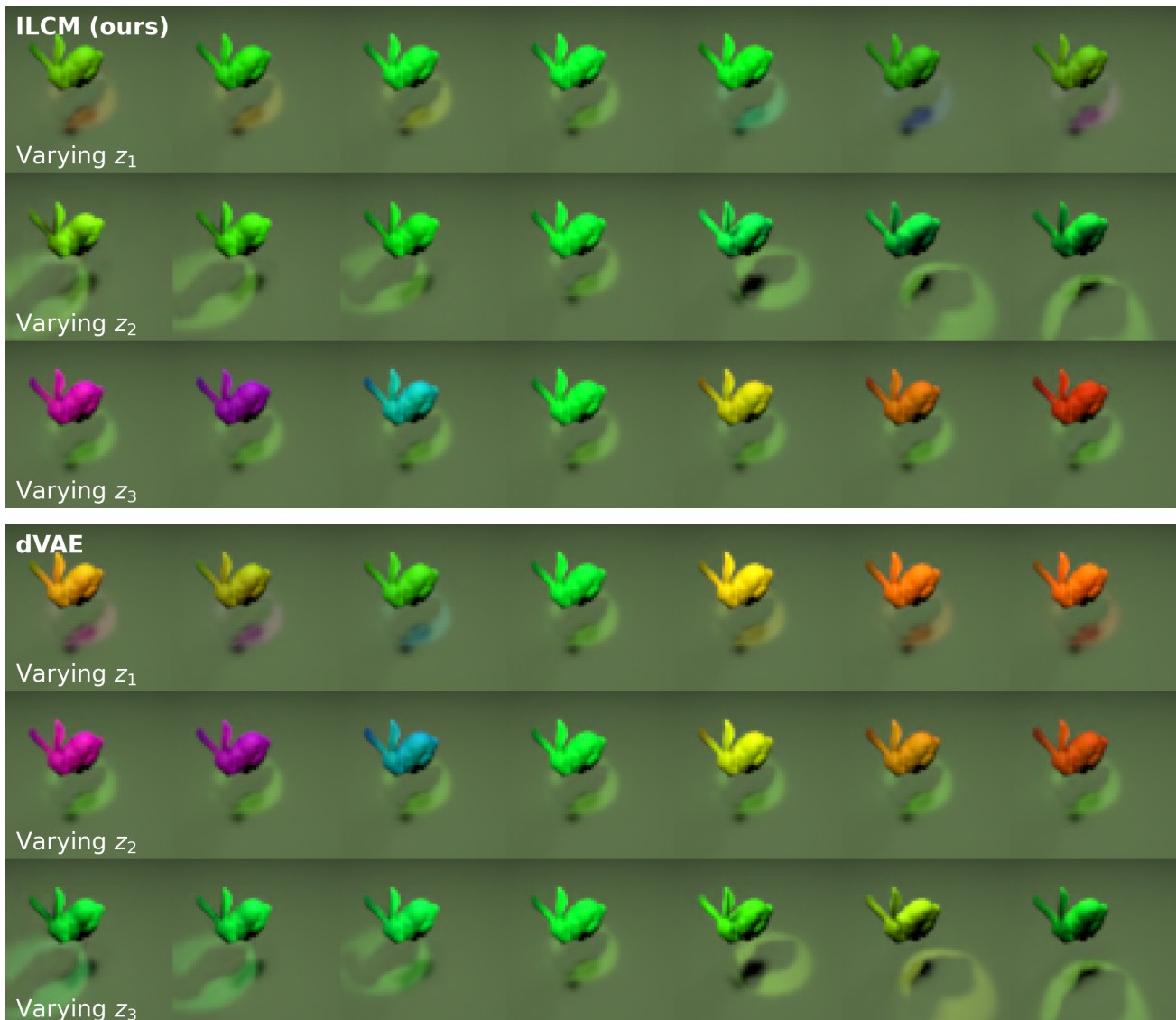

Figure 8: Effect of varying the learned causal factors on the image in the Causal3DIdent dataset. We encode a single test images (middle column) into the three learned causal variables. We then vary each of these causal factors in isolation (without performing interventions, that is, without including the causal effects on other variables) and show the reconstructed images. The ILCM (top) learns a representation that is quite disentangled: $z_1$ largely corresponds to the spotlight color, $z_2$ to the spotlight position, and $z_3$ to the object color. In contrast, the acausal dVAE baseline entangles the object color and spotlight color in its learned representation $z_1$.

the object hue, the spotlight hue, and the spotlight position. We use the following causal graphs:

- single edge: object hue → spotlight position;

- chain: spotlight position → spotlight hue → object hue;

- fork: spotlight hue → spotlight position, object hue;

- collider: spotlight hue → object hue ← spotlight position;

- full graph: spotlight hue → object hue → spotlight position, spotlight hue → spotlight position.

Since all of these properties are defined on a range $[0, 2\pi)$, we apply an elementwise $\mathrm{arctanh}$ transform and rescaling to our variables such that they populate a subset of $[0, 2\pi)$. This also avoids topological issues. Next, we generate images in $64 \times 64$ resolution following the procedure described in Ref. [14]. We use Blender [33] to generate 3D rendered images based on the previously defined causal variables. To increase diversity of the six datasets, we render each dataset with a different object: Teapot [34], Armadillo [35], Hare [36], Cow [37], Dragon [38], and Horse [39]. We generate $10^5$ training samples, $10^4$ validation samples, and $10^4$ evaluation samples.

**Architecture** For the noise encoder and noise decoder we use a convolutional architecture with four residual blocks, using downsampling via average-pooling and bilinear upsampling, respectively. We do not use BatchNorm, as we found that that can lead to practical issues when images in a batch are very similar. The output of the convolutional layers is then fed through a fully connected network with two hidden layers, 64 units each, and ReLU activations. For each of the three latents output by the encoder, we apply an additional elementwise MLP with one hidden layers, 16 units each, and ReLU activations. For the solution functions we use the same architecture as in the 2D toy data.

**Training** Models are trained using the procedure described in Sec. C.2. We train for $2.5 \cdot 10^5$ steps using a batch size of 64 and an initial learning rate of $8 \cdot 10^{-5}$. The weights of the different loss terms and regularizers are as follows: $\beta$ is initially set to 0 and increased to its final value of 0.05 during training, $\alpha = 10^{-2}$, and $\gamma = 5$ throughout training. For each method, we train models with three random seeds and in the end select the median run according to the validation loss.

**Results** In addition to the results shown in the main paper, we show metrics for each separate subset in in Tbl. 2. We also visualize the disentanglement properties of the learned representations in Fig. 8 and the ability of the ILCMs to infer and reproduce interventions in Fig. 9.

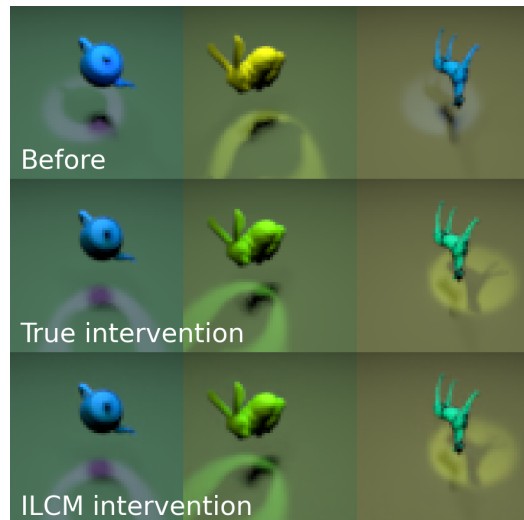

Figure 9: Causal3DIdent before (top row) and after (middle row) interventions, and post-intervention samples generated from the ILCM under the intervention inferred from the data, indicating we correctly learned to intervene.

### D.4 CAUSALCIRCUIT EXPERIMENTS

**Dataset** We introduce the new CausalCircuit dataset. This environment is built in the MuJoCo simulator [22], using a model of the TriFinger robotic platform [40], of which we use a single finger. There are four causal variables: the three lights and the robot arm. The arm state is the position along an arc that goes over three buttons. The position on the arc is translated into actuator angles using an inverse kinematic model. To obtain a sample, the arm is moved away from the buttons, then lowered to the desired position, after which a rendering is recorded. Each red, green and blue button then has a pressed state $b_R, b_G, b_B$ that depends on how far the button is touched from the center and scales linearly in the radial distance from 0 to 1. The causal model for the red, green and blue light variables and the arm $z_A$ then is:

$$v_R = 0.2 + 0.6 * \mathrm{clip}(z_G + z_B + b_R, 0, 1)$$
$$v_G = 0.2 + 0.6 * b_G$$
$$v_B = 0.2 + 0.6 * b_B$$
$$z_R \sim \mathrm{Beta}(5v_R, 5 * (1 - v_R))$$
$$z_G \sim \mathrm{Beta}(5v_G, 5 * (1 - v_G))$$
$$z_B \sim \mathrm{Beta}(5v_B, 5 * (1 - v_B))$$
$$z_A \sim \mathrm{Uniform}(0, 1)$$

The scene is rendered in $512 \times 512$ pixels using the MuJoCO renderer.

**Dimensionality reduction** Rather than training directly on images, we found it beneficial for fast experimentation to first condense the image datasets into a lower-dimensional representation and then train both ILCMs and baselines

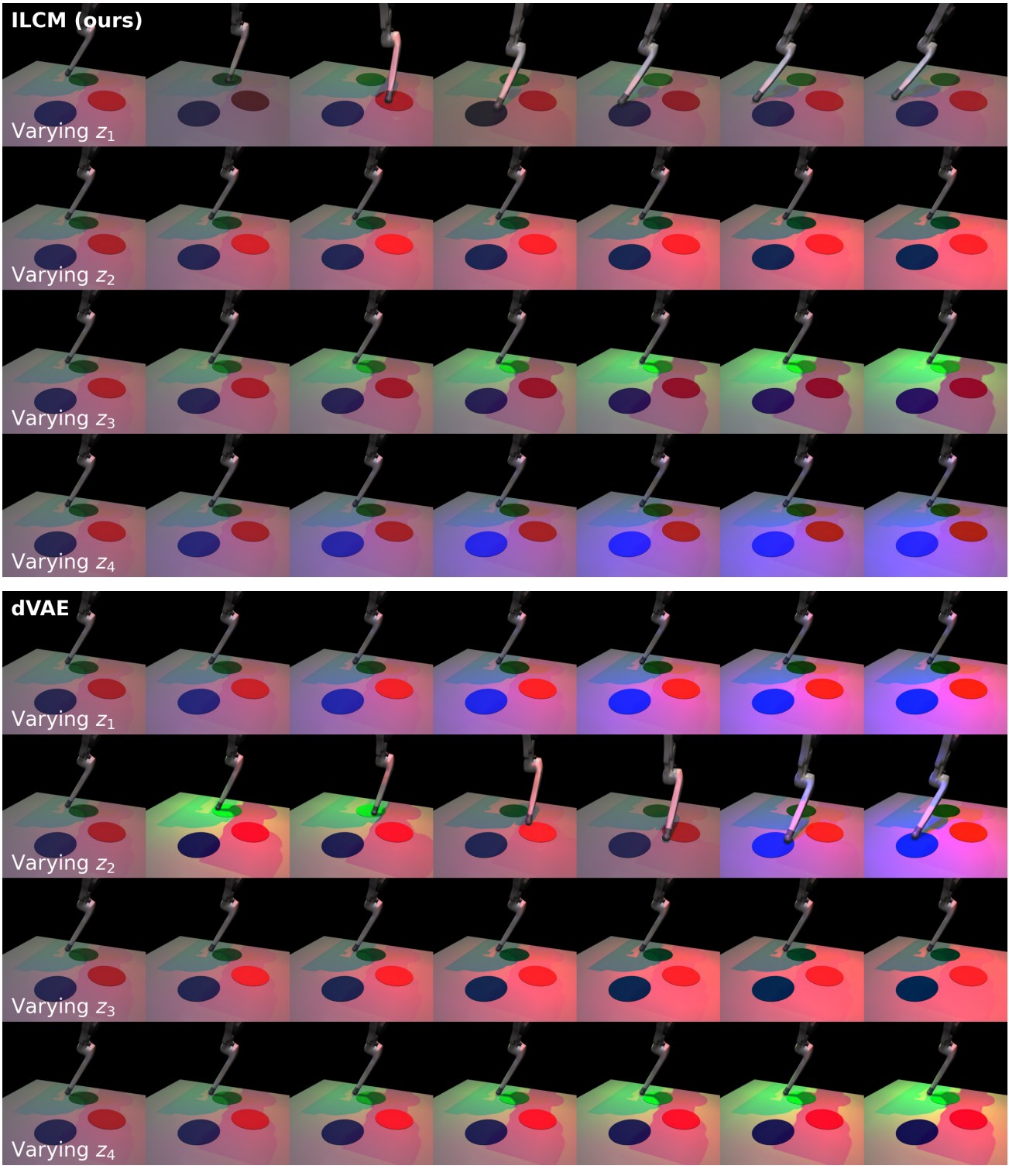

Figure 10: Effect of varying the learned causal factors on the image in the CausalCircuit dataset. We encode a single test images (left column) into the four learned causal variables. We then vary each of these causal factors in isolation (without performing interventions, that is, without including the causal effects on other variables) and show the reconstructed images. The ILCM (top) learns a representation that is quite disentangled: $z_1$ corresponds to the blue light, $z_2$ to the green light, $z_3$ to the robot arm position, and $z_4$ to the red light. In contrast, the acausal dVAE baseline entangles the different lights and the robot arm position in its learned latent factors.

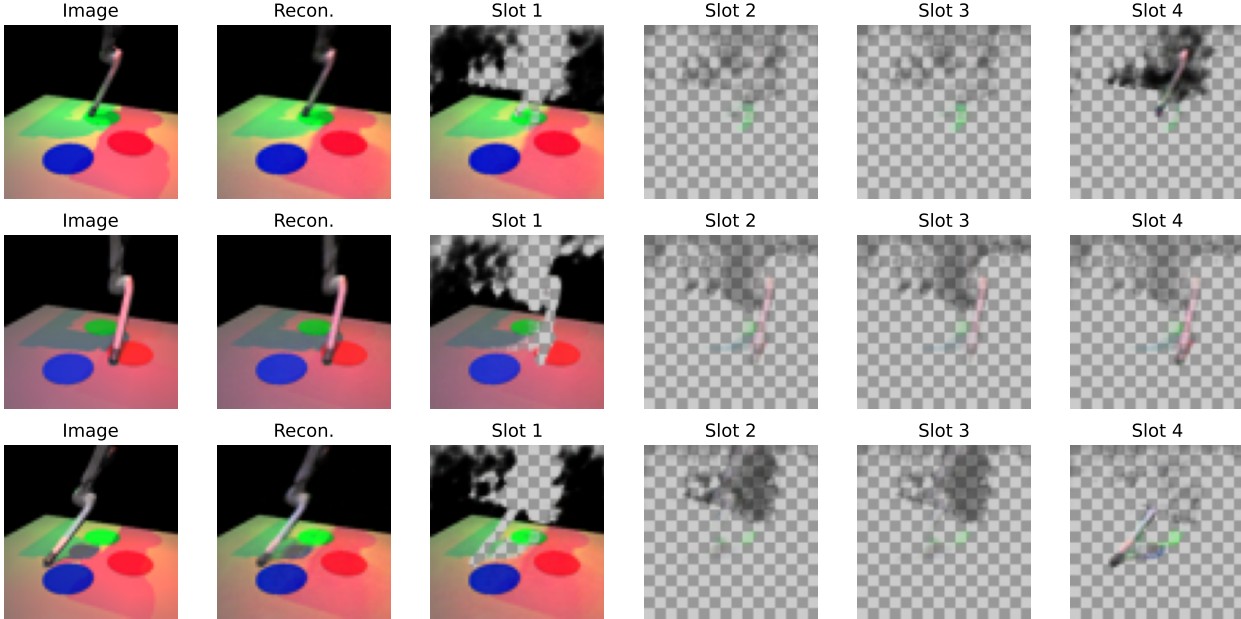

Figure 11: The slots found by slot attention. We see that the slots do not correspond to the causal variables. Only the arm is disentangled from the buttons and lights.

excluding slot attention) on that data. We used a $\beta$-VAE with a standard Gaussian prior with 16 latent dimensions; the encoder and decoder follow the same architecture as for the Causal3DIdent dataset. This VAE was trained for $8.5 \cdot 10^4$ steps using the Adam optimizer, an initial learning rate of $3 \cdot 10^{-4}$, cosine annealing, and batchsize 128.

For slot attention, we reduced the resolution to $64 \times 64$, as we found it difficult to train the model on the full resolution $512 \times 512$.

**Architecture** On this dimensionality-reduced, 16-dimensional data, we use fully connected networks for noise encoder and noise decoder, each with five hidden layers with 64 units each and ReLU activations. The solution functions have the same architecture as in the other experiments.

**Training** Models are trained using the procedure described in Sec. C.2. We train for $9.4 \cdot 10^4$ steps using a batch size of 64 and an initial learning rate of $3 \cdot 10^{-4}$. The weights of the different loss terms and regularizers are as follows: $\beta$ is initially set to 0 and increased to its final value of $3 \cdot 10^{-4}$ during training, $\alpha = 10^{-2}$, and $\gamma = 10$ throughout training. For each method, we train models with three random seeds and in the end select the best run according to the validation loss.

**Results** In addition to the results in the main paper, Fig. 10 shows that our ILCM model successfully disentangled the causal factors, while the dVAE baseline failed at that task. Similarly, in Fig. 11 we see that the slot attention model

fails to assign the causal variables into separate slots. We presume this is because the lights blend into each other, making them only describable by a single slot.

# E   EXPLICIT LATENT CAUSAL MODELS

## E.1   SETUP

Explicit latent causal models (ELCMs) are variational autoencoders, in which the latent variables are the causal variables of an SCM. They consist of a causal encoder $q(z|x)$, decoder $p(x|z)$, and a prior $p(z, \tilde{z})$ that encodes the causal structure.

Intervention targets can either be inferred with an intervention encoder $q(I|x, \tilde{x})$, similar to our ILCM model, or be marginalized over explicitly as $p(z, \tilde{z}) = \sum_I p(I)p(z, \tilde{z}|I)$. We have experimented with both settings, for concreteness we here focus on the second, simpler setup.

The conditional prior $p(z, \tilde{z}|I)$ is the weakly supervised distribution of an SCM. They are parameterized through the causal graph $\mathcal{G}$, neural causal mechanisms $f_i(\epsilon_i; z_{\mathbf{pa}_i})$, and noise base distributions. To learn the graph, the simplest option is to instantiate one LCM per graph equivalence class, train them, and select the model with the lowest validation loss. Alternatively, we can parameterize the graph in a differentiable way [18, 19, 41, 42] and learn the graph together with the other components through gradient descent.

We contrast the VAE setup of ILCMs and ELCMs in Tbl. 3.

Table 3: Differences between explicit and implicit latent causal models (ELCMs and ILCMs). Optional learnable components are shown in parantheses.

[1]For simplicity, in this section we describe an ELCM implementation without intervention encoder. [2]We achieved the best results when inferring and enforcing a topological order only for the last phase of ELCM training, see Sec. C.2.

|  | Explicit latent causal model | Implicit latent causal model |
| --- | --- | --- |
| Latent variables | causal variables $(z, \tilde{z})$
intervention targets $I$ | noise encodings $(e, \tilde{e})$
intervention targets $I$ |
| Learnable components | encoder $q(z\|x)$
decoder $p(x\|z)$
(intervention encoder $q(I\|x, \tilde{x}))^1$
graph $\mathcal{G}$
causal mechanisms $f_i(\epsilon_i; z_{\mathbf{pa}_i})$ | encoder $q(e\|x)$
decoder $p(x\|e)$
(intervention encoder $q(I\|x, \tilde{x})$)
(topological order)$^2$
solution functions $s_i(e_i; e_{\setminus i})$ |

Like ILCMs, ELCMs are trained on a VAE loss corresponding to a variational bound on $\log p(x, \tilde{x})$. Following common practice in causal discovery [18, 19, 43], we incentivize learning the sparsest graph compatible with the data distribution by adding a regularization term proportional to the number of edges in the graph to the loss.

### E.2 EXPERIMENTS

**Dataset** We experiment with ELCMs in similar datasets as we did in the main paper with ILCMs. In particular, we report results on six Causal3DIdent variations. However, we performed these datasets on an earlier iteration of these datasets: while main parameters of the scenes and the causal graphs are the same as in the experiments reported in the main paper, the ground-truth causal mechanisms are different. The metrics reported here are therefore not directly comparable to the ILCM results in the main paper.

**Hyperparameters** Our ELCM architecture and training follows similar hyperparameters to our ILCM experiments. We experimented with various graph parameterizations and sampling procedures, including directed edge existence probabilities with Gumbel-Softmax sampling [19, 41], undirected edge existence probabilities and edge orientation probabilities [18], and the parameterization through edge existence probabilities and a distribution over permutations [42]. While our implementation all of these methods were able to successfully learn causal graphs given the true causal variables, we were not able to reliably learn the representations and the graph jointly. We observed a higher success rate when training separate models for different fixed DAGs and then selecting the best graph based on the validation loss. The results reported below were generated with this exhaustive graph search strategy. Again we show the median run out of three random seeds according to the validation loss.

**Results** In Tbl. 4 we show disentanglement scores and learned graphs. The results are mixed: in some of the

Table 4: ELCM experiments on Causal3DIdent datasets. We show the learned causal graph, the structural Hamming distance SHD between the learned and the true graph and the DCI disentanglement score ($D$). The datasets differ slightly from the ones used in our main experiments, so metrics are not directly comparable.

| True graph | $D$ | Learned graph | SHD |
| --- | --- | --- | --- |
|  | 1.00 |  | 0 |
|  | 0.99 |  | 0 |
|  | 0.45 |  | 3 |
|  | 0.98 |  | 0 |
|  | 0.98 |  | 0 |
|  | 0.43 |  | 2 |

datasets the graph was correctly identified and the variables are disentangled, while in others the model failed at both tasks. Notably, we find that the results strongly vary with the initialization (i. e. the random seed). In Tbl. 4 we only show the median result out of three runs, but in almost all datasets there is one random seed that lead to successful disentanglement (always with the best training and validation loss) and one random seed that led to failed disentanglement (with a worse training and validation loss). ELCM training is thus much less robust than our ILCM experiments, where the results were largely stable across random seeds.

**Discussion** This result hints at the presence of local minima in the loss landscape that models can get stuck in when starting from an unlucky initialization. By manually analyzing the trained ELCMs, we find that one common failure mode us that models learn variables that are to some extent disentangled and the graph has the right skeleton, but some of the causal effects are wrongly oriented (swapping cause and effect). Such graphs are often in the same Markov equivalence class as the correct graph, which is why such a model can minimize the observational contribution $-\log p(z)$ to the overall loss. Smoothly changing the representations would take the model out of the Markov equiva-

lence class, increase this term and thus the overall loss; this configuration thus presents a local loss minimum. The same phenomenon occurs in our experiments with differentiable graph parameterizations.

## F  POTENTIAL SOCIETAL IMPACT

Although we expect the immediate societal impact of this work to be negligible, more generally causal representation learning may have significant impact in the longer run. It will allow for the discovery of potential causal relationships in unstructured human-centric data. This may be beneficial, for example as it allows one to inspect if a model has learned sensible or fair causal relationships. A potential risk is that, for example because certain confounding variables are not discovered from the data, the algorithm may conclude erroneously that sensitive variables are causes of relevant outcomes. Users of the algorithms should be cautious of that.