# OpenReview forum: "Weakly supervised causal representation learning"
_auai.org/UAI/2022/Workshop/CRL — CRL@UAI 2022 Poster_

### Official Review · Reviewer_U1i1 · 2022-06-25
**Review of "Weakly supervised causal representation learning"**

**Rating:** 7
**Confidence:** 4

**Review:**

## Summary:

This work studies the fundamental problem of causal representation learning. In the most general case, it's well known that given raw data, this task of recovering the true generative model is impossible because of lack of identifiability. To get around this, we either make assumptions about the generating process or obtain more information such as by observing auxiliary variables. In this work, they assume that we are given both the raw data and the data after it has been intervened by "atomic" interventions (those that act on only one variable). The targets of the interventions are not needed, only the intervened data. Under this assumption, they theoretically argue that the latent generative model is identifiabile. Moreover, they test their theory via experiments on a toy dataset and two image datasets, one of which they construct based on robotic arm manipulations. This work is a core contribution in the study of when identifiability holds for latent causal models. Moreover, the learnt models have various advantages for downstream tasks, that are yet to be explored in future works. The proposed setting of weak supervision via intervention is novel (in this context) to the best of my knowledge and the experiments are convincing. I have verified as much as possible and the theory and proofs seem sound. Finally, the paper is well-written, rigorous and the work also raises many interesting research directions, therefore I recommend acceptance.

## Technical summary and strengths:

In more detail, the model is assumed to be generated by a latent causal model: Here the noise variables \eps_1, ..\eps_n are independently sampled and they are used to generate causal variables z_1, .., z_n using a nonlinear DAG model. Finally, these causal variables generate the observed data x via a nonlinear map. An intervention is when \eps_i and z_i are resampled, independent of z_i's parents in the DAG. This work assumes that intervened data \tilde{x} is observed along with x, more precisely, pairs (x, \tilde{x}) are observed. Under various assumptions, this model is shown to be theoretically identifiable upto pointwise diffeomorphisms and permutations of the variables. This result can be thought of as extending the work of Locatello et al. 2020, that the authors point out.

Main technical assumptions are
- The maps that map from noise \eps to causal variables z, as well as the map from z to x are assumed to be diffeomorphisms
- All interventions are assumed to be atomic, i.e. they act on at most one target pair (\eps_i, z_i).
- Interventions are perfect, i.e. intervention on index i, does not depend on the parents of i.
- All possible interventions are observed (i.e. the support of the set of interventions contains all indices i)
- All noise variables and causal variables are assumed to be real numbers.

Appendix B does a good job of explaining the necessity and restrictivity of these assumptions and suggest directions for future work, depending on which assumptions have potential to be relaxed. Intuitively, the main identifiability result makes sense because all interventions are observed, so there is not much wiggle room for the latent model (since we also assume diffeomorphic maps). But making this precise is fairly nontrivial and the authors provide a neat category-theoretic proof of their main theorem. While strictly not necessary, the notion of string diagrams make the proof easier to follow, as opposed to a bare-bones proof that could be tedious.

To augment the theory, the work runs experiments on a 2 dimensional toy dataset and two image datasets with 3 or 4 causal variables. After learning the generative model via standard VAE type approaches (with an augmented ELBO loss), they extract the underlying DAG either via using blackbox algorithms from causal inference or a heuristic they propose that is based on order search, giving rise to two kinds of models.

To measure the performance of their model, they report the disentaglement score, intervention accuracy and the structural hamming distance metric. Because the setting is novel, there isn't much prior work to compare to but they are able to adapt some models from prior works (while making some unrealistic assumptions like independence of the interventions). Their experiments perform quite well and the metrics are promising.

## Weaknesses:

It's unrealistic to expect the entire collection of assumptions to hold in some real-data task. While its possible to ameliorate some of the assumptions by minor fixes (as outlined in appendix B), the applicability of the main theorem is somewhat limited. Therefore, the contribution in the current work is primarily theoretical and conceptual.

The models in the experiments are not exactly a plug-and-play of the theoretical ideas. There was significant effort spent in making the models work well on the datasets. This includes hyperparameter tuning as well as several heuristics. For instance, they had to appropriately modify the ELBO loss by tuning the significance of the reconstruction error (just as in beta-VAEs), and also add two extra regularizers with appropriate scaling whose hyperparameters have to be tuned. This modification is done in order to prevent posterior collapse (which is a well-known phenomenon in VAE training). The training proceeds in 4 phases, with a pretraining done first, followed by training without the solution functions, followed by training with the solution functions, and finally followed by the graphical structure learning. There are other technical heuristics such as freezing the layers and making the intervention encoder determinnistic. This illustrates that making the experiments work is not straightforward and can be a lot of work. Moreover, as the authors pointed out, the experiments are not scalable. That said, its well known that theory and practice in causal representation learning are different beasts so this is somewhat expected. Therefore, it's neat that they are able to make the entire pipeline work well.

Unless I'm missing something, the number of causal variables is assumed to be known beforehand. While this is not a significant assumption, this should be explicitly clarified.

## Other comments:

Does the theory go through if we replace the class of diffeomorphisms by an appropriate subclass?

In page 4, it's super hard to distinguish \bar{z_i} and \tilde{z_i} and this left me confused for a while. I'd recommend using some other notation.

In page 2, "SCM by replacing for a subset" --> "SCM by replacing a subset"

In page 13, the 4th display equation for tilde(z_{o'}) for the left part overflows the column

In page 18, "a feedback the randomly" --> "a feedback of the randomly"

---

### Official Review · Reviewer_Y7wW · 2022-07-01
**good work with clear contributions**

**Rating:** 6
**Confidence:** 4

**Review:**

Summary:
This paper provides identifiability guarantees for causal variables and their causal structures by observing the effect of interventions in a weakly supervised setting. An implicit latent causal model based on the variational encoder is introduced to tackle the problem. Moreover, the experimental results show the effect of the method.


Strengths:
The motivation is clear. The work is novel. The identification for both the causal representations and the structural causal model between them is an interesting problem in a weakly supervised setting.

The presentation is clear and easy to follow. The authors introduce the problem setup clearly and explain the proposed method in detail.


Weaknesses:
The authors are suggested to compare the technical difference between yours, the work of Locatello et al. [5]  and some other related prior work to clarify the technical contribution.

In the experiment, the causal graphs are quite simple with only two or three or four variables. The effectiveness of the method could be shown with more experiments with any size of variables.

As the authors say, the method only works reliably with continuous causal variables, for which, some analysis would be better to provided.

In the discussion, it would be better to provide some reference for the motivation of the setting: “learning causal structure from passive observations of an agent (or demonstrator) interacting with a causal system.”

---

### Meta-Review · Program_Chairs · 2022-07-06

**Recommendation:** Accept (Poster)
**Confidence:** 4

**Metareview:**

Great paper, should be accepted at the workshop

---

### Decision · Program_Chairs · 2022-07-06

Accept (Poster)